# IDRBench: Understanding the Capability of Large Language Models on Interdisciplinary Research

Yuanhao Shen [* 1]   Daniel Xavier de Sousa [* 2]   Ricardo Marçal de Andrade Nascimento [2]
Hongyu Guo [3]   Xiaodan Zhu [1]

## Abstract

Innovation is a key driving force of human civilization. As the body of knowledge has grown considerably, bridging knowledge across different disciplines, where significant innovation often emerges, has become increasingly challenging. The recent advancements in machine learning models, particularly Large Language Models (LLMs), have provided effective access to extensive knowledge sources and shown impressive abilities in reasoning, rendering significant opportunities for interdisciplinary discovery. Our research aims to understand the capabilities of state-of-the-art LLMs to integrate knowledge from different fields for interdisciplinary research (IDR). To address this fundamental problem, we introduce IDRBench, a pioneering framework that includes both datasets and evaluation tasks: (1) IDR Paper Identification, (2) IDR Idea Integration, and (3) IDR Idea Recommendation. Our study of ten mainstream LLMs provides a comprehensive analysis of their behavior and establishes benchmarks and baselines for future research. To the best of our knowledge, IDRBench is the first to provide a comprehensive investigation of LLMs' IDR capabilities. Our data and code are available at ⭘ Github and 🤗 Hugging Face.

## 1. Introduction

*"If you would understand anything, observe its beginning and its development."*

*—Aristotle*

Aristotle, one of history's greatest polymaths, made en-

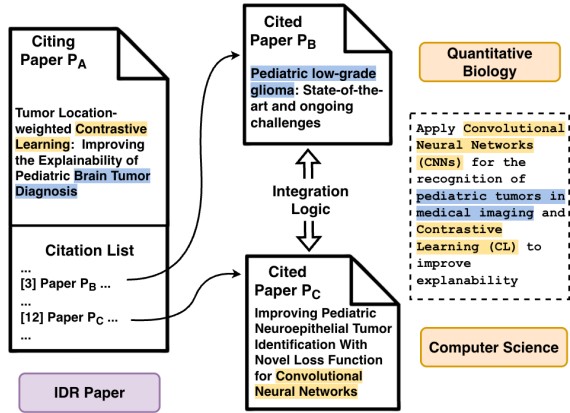

*Figure 1.* Data format in IDRBench. The key notions (in highlighted text) from cited papers $P_B$ and $P_C$ are integrated (more than merely referenced) to generate the IDR Paper $P_A$. A complete positive paper triplet also includes the specific integration logic (in dashed box) from $P_A$.

during contributions across philosophy, biology, physics, logic, and more areas, shaping centuries of intellectual development. However, as the body of knowledge has grown considerably, modern science has fragmented into more specialized areas, making it increasingly difficult for humans to digest and integrate knowledge from different domains to make interdisciplinary innovation. At the same time, the recent significant advancements in Large Language Models (LLMs) (OpenAI et al., 2024; OpenAI, 2024; Grattafiori et al., 2024; Anthropic, 2025; DeepSeek-AI et al., 2025), with their improved reasoning abilities and access to massive corpora spanning diverse scientific fields, have shown significant potential in scientific discovery. Frameworks that endow LLMs with nascent abilities to assist research ideation (Shi, 2024; Lu et al., 2024), propose research ideas (Si et al., 2025), and deploy research code (Bogin et al., 2024) have catalyzed the emergence of efforts toward measuring LLMs' ideation ability (Guo et al., 2024; Lin et al., 2024; Liu et al., 2025b; Yang et al., 2025).

Despite these recent advancements, there is a notable gap in understanding the basic capabilities of LLMs to digest and integrate knowledge from different domains for interdisciplinary discoveries, leaving scarce evidence to understand

---

[*]Equal contribution  [1]Department of Electrical and Computer Engineering & Ingenuity Labs Research Institute, Queen's University, Canada [2]Instituto Federal de Goiás, Anápolis, Brazil [3]National Research Council Canada. Correspondence to: Yuanhao Shen <23rq31@queensu.ca>.

*Proceedings of the 43rd International Conference on Machine Learning*, Seoul, South Korea. PMLR 306, 2026. Copyright 2026 by the author(s).

the key research question: *Are LLMs capable of conducting interdisciplinary research?*

Answering this question is not easy. To the best of our knowledge, none of the existing works have assessed this capability of the state-of-the-art LLMs. One major issue that hinders such evaluation is the lack of a dedicated benchmark specifically designed for interdisciplinary research (IDR). The absence of a benchmark also makes it more difficult to measure the actual improvements brought by new or future approaches when they are compared to the baselines, increasing the risk of misleading comparisons and reducing the reproducibility of IDR findings.

We introduce IDRBench, a novel framework that includes both datasets and evaluation tasks, based on scientific publications sourced from the ArXiv platform covering six distinct disciplines. Our design of the evaluation tasks follows a progressive, real-world perspective, reflecting the stages of interdisciplinary research idea development, including (1) *IDR Paper Identification*, (2) *IDR Idea Integration*, and (3) *IDR Idea Recommendation*. The IDRBench dataset is structured as triplets as illustrated in Figure 1. For instance, an IDR paper titled "Tumor Location-weighted Contrastive Learning: Improving the Explainability of Pediatric Brain Tumor Diagnosis" serves as "Citing Paper $P_A$", accompanied by at least two "Cited Papers" $P_B$ and $P_C$ from the disciplines of quantitative biology and computer science, respectively. We conducted evaluation on 10 mainstream LLMs, providing the first comprehensive study of how the state-of-the-art models behave, and establishing baselines for future studies. Our key contributions are threefold:

- We introduce the first comprehensive, closed-book benchmark IDRBench, designed to evaluate LLMs on interdisciplinary ideation. IDRBench includes a high-quality human-annotated dataset as well as different yet specialized IDR tasks.

- We present the performance of state-of-the-art LLMs and their behavior. Our analysis shows that LLMs are capable of generating valid and useful ideas as verified by human experts. However, LLMs still struggle to reliably distinguish true interdisciplinary integration, and the reasoning-oriented models could degrade IDR performance.

- With preliminary explorations of downstream applications, we show how IDRBench can potentially serve as a standard testbed and support broader research in interdisciplinary discovery and evaluation.

## 2. Related Works

**Using LLMs for IDR.** Some recent research efforts have started to explore the effective use of LLMs for assisting interdisciplinary research. For example, Zheng et al. (2024) proposes Disciplink, a framework that assists researchers in seeking and gathering potential IDR topics. Similarly, Liu et al. (2025a) leverage LLM persona simulation with human-in-the-loop integration to enhance IDR topic formulation. A number of NLP and machine learning techniques, including summary-text generation (Čeović et al., 2023; Li et al., 2024), sample-efficient learning (Gharoun et al., 2024), and content-aware analysis (Van Der Meer, 2016) are utilized to conduct IDR paper classification (Xu et al., 2024; Liu et al., 2024). None of these works, however, were conducted to study and evaluate the capabilities of LLMs for IDR.

**LLMs' Research Ideation Ability.** More relevant to our work, Lin et al. (2024) introduce SchNovel along with a retrieval-augmented framework to assess the novelty of proposed research ideas. The works proposed in (Guo et al., 2024; Liu et al., 2025b; Yang et al., 2025; 2024) take the perspective of hypothesis generation to curate datasets that evaluate LLMs on formulating novel hypotheses. However, none of these works focuses on IDR. By contrast, studies such as (Liu et al., 2024; Xu et al., 2024; Liu et al., 2025a; Zheng et al., 2024) merely perform survey-based studies. We summarize the key differences between IDRBench and existing works in Table 1. Unlike the prior work, IDRBench provides evaluation data and tasks for interdisciplinary research (as shown in the IDR column). The second column indicates whether a benchmark contains multiple tasks that enable progressive evaluation with increasing complexity (see Section 3 for our detailed subtasks). The *source* column indicates the origin of each dataset. In addition to the above differences, IDRBench also offers a larger dataset compared to existing ones, as shown in the number-of-papers column.

## 3. The IDRBench

In our study, we adopt the widely accepted definition of interdisciplinary research (Cantone, 2024; National Research Council, 2014; Nakhoda et al., 2023; Sell et al., 2022) as: *"**integrate** information, data, techniques, tools, perspectives, concepts, and/or theories from two or more disciplines or bodies of specialized knowledge to advance fundamental understanding or to solve problems whose solutions are beyond the scope of a single discipline or area of research practice."* Inherently, IDR can be considered a creative task, as integrating knowledge from two or more disciplines fundamentally requires drawing insightful analogies across distinct domains, which, from the perspective of Hofstadter & Sander (2013), lies at the core of human thinking.

Our proposed IDRBench, a pioneering framework to assess the capabilities of LLMs for IDR, consists of two key components: (i) *The IDRBench Dataset*: a benchmark dataset sourced from six disciplines, and (ii) *The IDRBench Task Set*: a suite of progressive evaluation tasks that capture the capabilities of LLMs for IDR idea identification, ideation, and recommendation.

| Dataset | IDR | Progressive Evaluation | Source | Data Type | # Papers | Data Format |
|---------|-----|------------------------|--------|-----------|----------|-------------|
| **IDRBench (Ours)** | ✓ | ✓ | ArXiv | Annotated Synthetic | 1,014 31,437 | Structured paper triplet $[P_A; (P_B, P_C)]$ |
| SchNovel(Lin et al., 2024) | ✗ | ✗ | ArXiv | Annotated Synthetic | – 15,000 | Paper similarity score |
| IdeaBench(Guo et al., 2024) | ✗ | ✗ | Semantic Scholar | Annotated Synthetic | – 2,374 | Biomedical paper + Reference list |
| ResearchBench(Liu et al., 2025b) | ✗ | ✓ | Semantic Scholar + CrossRef | Annotated Synthetic | 588 798 | Paper + Reference list |

*Table 1.* Comparison of IDRBench with related works. IDRBench includes three tasks, as detailed in Section 3.

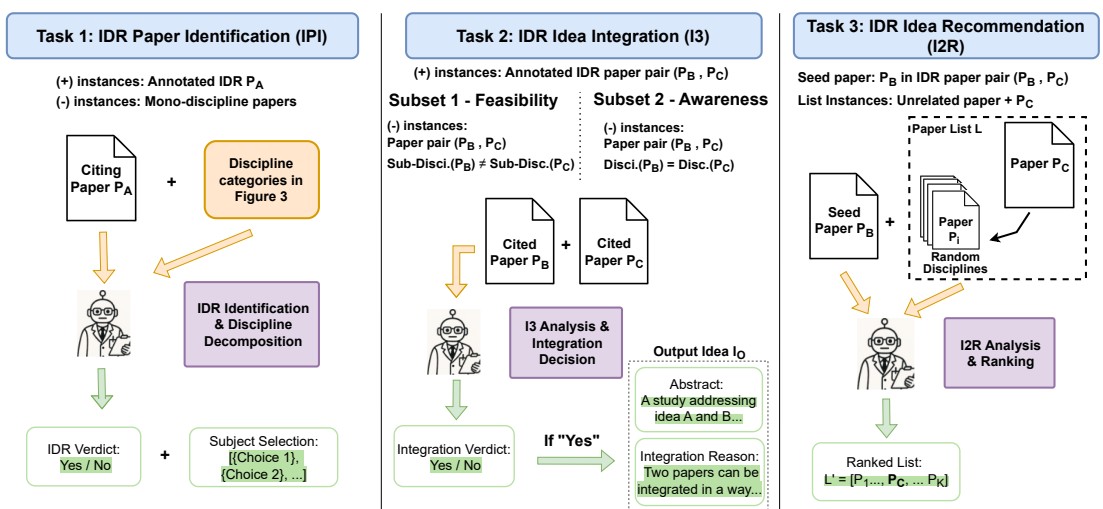

*Figure 2.* Visualization of tasks IPI, I3, and I2R and their dedicated datasets within IDRBench. Orange and green arrows stand for input and output flow, respectively. Purple boxes summarize the LLM's functions. Disci. stands for the abbreviation of Discipline.

## 3.1. The IDRBench Dataset

To address the challenge of identifying interdisciplinary research, which is often nuanced, we leverage the insights from (Zhu et al., 2015) that a paper's core innovation typically stems from a leading idea within its citation clusters. We hereby extend this notion and consider that one can differentiate an IDR paper that is on the more clear-cut regions of the IDR spectrum based on the integration of key citation papers. Thus, to effectively represent citation-based IDR, we introduce a knowledge triplet illustrated in Figure 1. For instance, an IDR paper serves as "Citing Paper $P_A$", accompanied by two "Cited Papers" $P_B$ and $P_C$ from the disciplines of *Biology* and *Computer Science*. An IDR triplet is represented in the form $[P_A; (P_B, P_C)]$ and can be generalized to include more than two cited papers[1]. By utilizing the proposed triplets, we enable a more versatile decomposition when adapting to the various evaluation tasks detailed in Section 3.2.

**Data Preparation.** We create our IDRBench from the ArXiv platform[2], containing 271,348 papers uploaded between November 2024 and October 2025. Our choice to use ArXiv is motivated by two main reasons: (i) many of the recent papers uploaded to ArXiv have not yet been published in any formal venue, which reduces the risk of information leakage during LLM pretraining (similar problem is also noted in (Liu et al., 2025b)). This can be further mitigated by using our data collection pipeline for new dataset construction, available in Appendix A.1. (ii) ArXiv requires authors to select specific category taxonomies when submitting papers, including disciplines and sub-disciplines. Therefore, (sub)discipline categories labeled by the authors serve as the preliminary criteria to filter out the potential IDR and non-IDR papers[3].

To further enhance the level of interdisciplinarity of our outcome dataset and reduce the possibility of nuanced IDR samples, we combine the disciplines in ArXiv categories

---

[1]Although in Section 3 we show that 93% of IDR papers associated with more than one discipline are described by the authors as a combination of only two primary cited papers.

[2]https://arxiv.org/

[3]Although ArXiv has discipline labels, the final selection of an IDR paper is done by our expert annotators.

that are deemed less distant from each other. For example, creating a new paper that combines *Computer Science* and *Electrical Engineering* is more common and straightforward than integrating *Computer Science* and *Biology*, where the two areas are deemed to be more distant. To mitigate this issue, we manipulate the ArXiv's categorization by selecting specific combinations of disciplines that are conceptually more distant from one another, and merging closely related ones into single categories to reflect their conceptual proximity. For instance, we merged *Computer Science, Electrical Engineering, and System Science* into a single discipline. Consequently, the associated sub-disciplines were also integrated – the categorization in this work is described in Figure 3.

**Positive Instances:** Using the preprocessed dataset, we collected 335 paper triplets containing more than 1,000 papers via human annotation to build the positive instances[4]. For each data sample, human experts have access to the full content of the paper. To collect a positive paper triplet, the experts must first confirm that paper $P_A$ is an IDR. Then, the experts examine the reference list of $P_A$ (with an average length of 35 papers) to identify the $P_B$ and $P_C$ papers from different disciplines. Although annotators may select papers from up to four disciplines (at least one paper for each discipline) in our web platform, we observe that almost 95% of papers $P_A$ describe the integration between only two disciplines. Moreover, when evaluating the ArXiv Platform, we also note that among all papers associated with more than one discipline (359,937 papers), 93% integrate only two of them. Both the observations from our annotators and the ArXiv data statistics reinforce our assumption that a valid IDR idea is most commonly derived from the integration of two distinct disciplines.

To ensure the quality of the 335 positive samples and trace their publication footprint, we note that 80.2% of the preprints have been published or are currently under review by academic conferences or workshops, while 63.6% of the unpublished preprints have already received citations. This further confirms that arXiv captures novel research that has the potential to mature into formal publications.

Figure 3 describes our positive data distribution across the combined discipline categories. From all categories in ArXiv, and based on the outcome of our expert annotation, the figure shows that our distribution includes a substantial portion of more distant discipline combinations. For example, the integration of ideas between [*"Quantitative Biology", "Physics", and "Other"*] and *"Computer Science, Electrical Engineering and Systems Science"* represents nearly 55% of our annotated dataset. We use *"Others"* to

---

[4]Experts were paid to annotate the positive paper triplets at an hourly rate higher than the minimum wage in the location where the annotation and research were conducted.

refer to disciplines identified by the annotators that are not included in the ArXiv taxonomy, such as *Medicine, Chemistry, Law, and others*.

To further ensure the quality of the positive samples, we impose a mutual-agreement procedure among experts for the positive samples to validate that the paper triplet elements indeed represent the main works from the disciplines being combined. Each paper triplet is independently annotated by two experts, and we include only instances in which both annotators mark the sample as positive. Further details regarding the annotation process are provided in Appendix A.1.

**Data Collection Pipeline:** We describe in Appendix A.1 our pipeline for quick data collection and annotation as well as the curation of synthetic data, which could be continuously used to evaluate future LLMs, if data contamination is a concern. In this paper, we run experiments and report our main results using the expert-annotated data from the pipeline without relying on synthetic data. We include the details of the pipeline in Appendix A.1 and the discussion of synthetic data in Appendix A.8.

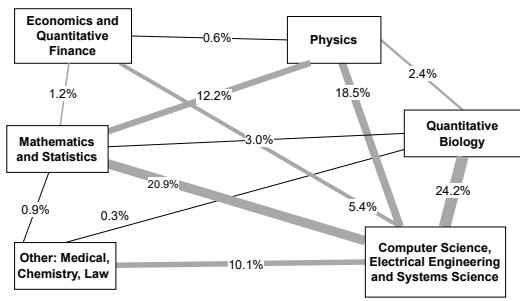

*Figure 3.* Demographics of discipline combinations for 335 positive samples in IDRBench that are annotated by human experts.

### 3.2. The IDRBench Tasks

Taking advantage of the versatility of the paper triplet structure, we design three tasks to evaluate LLMs' inherent IDR capabilities. Figure 2 provides an overview of the tasks in IDRBench, which include *(i) IDR Paper Identification (IPI)*, evaluating LLMs' ability to classify IDR papers; *(ii) IDR Idea Integration (I3)*, testing LLMs' ability to combine papers from distinct disciplines into a feasible and novel IDR idea; and *(iii) IDR Idea Recommendation (I2R)*, evaluating LLM's ability to recommend the most appropriate IDR paper given a list of unrelated papers. The design of the three tasks focuses on different perspectives when conducting interdisciplinary research in a progressive manner, while maintaining the interpretability of the model behavior.

### 3.2.1. IDR PAPER IDENTIFICATION (IPI)

**IPI Task Description.** The pipeline for **IDR P**aper **I**dentification (IPI), shown in the first column of Figure 2, measures the fundamental ability of LLMs to identify whether a scientific paper is an IDR. Given the title and abstract of a citing paper $P_A$, the LLM is first instructed to perform classification on whether $P_A$ is an IDR from at least two distinct disciplines. In the case of a positive classification, it is further instructed to refer to the discipline categories derived from Figure 3 and decompose $P_A$ into a combination of discipline choices.

**Dataset in IPI.** We build the IPI dataset with positive and negative examples, where each instance is a description of paper $P_A$ (title and abstract), a label (yes or no), and (in the case of yes) two disciplines integrated into $P_A$ (from the proposed triplet; in this task, we only use $P_A$). The positive instances are derived from human-annotation results, considering 335 papers. For negative instances, we select from ArXiv papers where **authors include only one sub-discipline and one discipline in their submissions**. In fact, although negative papers may offer novel contributions, their scope is confined to a single discipline without interdisciplinary overlap.

We have observed from the whole ArXiv dataset (2,477,207 papers) that 85% of papers have only one discipline assigned by the authors, which also aligns with previous findings (Boyack et al., 2005), noting that IDR connections are less frequent and sparser than those in non-IDR papers. Thus, we set an approximate 1:10 ratio between positive and negative samples to reflect the nature of IDR research and ensure a realistic evaluation setup. This assumption of sparsity is also used throughout the subsequent tasks.

### 3.2.2. IDR IDEA INTEGRATION (I3)

**Task Description.** IDR **I**dea **I**ntegration (I3), showcased in the second column in Figure 2, takes a step further to investigate the IDR integration abilities of LLMs. We provide LLMs with two cited papers $P_B$ and $P_C$ that contain the title and abstract. The LLM is thereafter instructed to perform an idea integration analysis under an IDR setting, predicting whether the integration between $P_B$ and $P_C$ is a promising IDR idea. The integration analysis focuses on three criteria, including (1) whether the potential outcome idea integration aligns with the official definition of IDR, (2) whether the integration of the two papers yields a feasible IDR, and (3) whether the output IDR idea bears sufficient novelty. We assume that the combination of $P_B$ and $P_C$ is only viable when all three criteria are met. If the answer is yes, we further prompt the model to generate an IDR idea output, represented by a working-paper abstract and integration reasoning using $P_B$ and $P_C$. The reasoning output facilitates our understanding and provides informative insights during the idea integration stage of IDR. We note that in the evaluation, only $(P_B, P_C)$ pairs associated with the positive $P_A$ samples are considered ground truth.

To sufficiently evaluate whether LLMs are truly identifying IDR, one needs to ensure that papers $P_B$ and $P_C$ exhibit a meaningful and reliable distance between their disciplines and sub-disciplines. To address this challenge, we construct two data subsets for evaluating feasibility (Dataset 1) and awareness (Dataset 2).

**Dataset 1 in I3.** For positive samples, we use the $(P_B, P_C)$ pairs that are associated with the annotated IDR paper $P_A$. For the negative samples in Dataset 1, we measure how reliable LLMs are in providing feasible combinations. Specifically, two papers are randomly selected from two **distinct disciplines** (following Figure 3), decreasing the likelihood of selecting a feasible IDR idea. This approach is based on the above observation that valid interdisciplinary ideas are extremely sparse within the overall space of possible paper combinations.[5] Therefore, considering negatives from the undirected aspect of randomness, i.e., as long as two random papers are selected from two random yet distinct disciplines (and naturally distinct sub-disciplines), they are considered a negative pair. In Dataset 1, we also use a 1:10 ratio between annotated positive and curated negative pairs.

To quantify the reliability of our negative sampling strategy, we conducted a targeted estimation of the False Negative Rate (FNR) for I3 Dataset 1. We randomly sampled 70 negative pairs and asked 3 independent human evaluators to assess whether each pair could plausibly form a valid IDR combination. With an inter-annotator agreement rate of 85.5%, the estimated FNR was 5.8% (considering ratings of 4 and above on a 5-point scale as false negatives). The low FNR supports the validity of our random negative sampling strategy and is consistent with the well-documented sparsity of genuine IDR connections in the scientific literature (Boyack et al., 2005).

**Dataset 2 in I3.** For positive pairs, we follow the same Dataset 1 settings. However, for negative pairs, we introduce a semi-random selection method to measure LLMs' awareness of IDR. We now consider that two papers randomly selected from **the same discipline and sub-discipline** are negative instances. Thus, from a positive pair $P_B$ and $P_C$, we replace $P_B$ with a randomly selected paper $P_D$ from the same sub-discipline as $P_C$. Although sampling pairs from the same sub-discipline has a higher chance of forming a valid research idea, it does not follow the IDR definition,

---

[5]We emphasize that selecting negative instances from a random strategy is not necessarily new. In information retrieval for instance, recommender systems works (Rashed et al., 2021; He et al., 2017; Elkahky et al., 2015) usually rely on random sampling due to the sparse space of non-relevant items in training data.

and the samples are thus considered negative. For Dataset 2, we maintain an approximate 1:10 ratio between annotated positive and curated negative pairs.

To sum up, in Dataset 1 we blend positive and negative pairs from different disciplines. By doing this we are examining the LLMs ability to evaluate feasible IDR ideas, since randomly selecting papers from different fields should not constitute an IDR idea. On the other hand, in Dataset 2 we blend positive instances using pairs from different disciplines and negative pairs from the same discipline and sub-discipline, aiming to assess LLMs' awareness of properly constraining their scope in the context of IDR. Besides, the two subsets together offer complementary evaluation perspectives that are instructive for future IDR frameworks: subset 1 adopts a more permissive regime that allows a few false negatives, whereas subset 2 enforces a more conservative setting, penalizing false negatives more strongly. We showcase the composition of the two datasets in the second column of Figure 2.

### 3.2.3. IDR IDEA RECOMMENDATION (I2R)

**Task Description.** IDR Idea Recommendation (I2R) is the most complex task in IDRBench, since it tests the LLMs' holistic IDR ability in a multiple-round recommendation setting: given a cited paper pair $P_B$ and $P_C$ from a positive $P_A$, the LLM receives $P_B$ as the seed paper and a candidate paper list $L$, while the other paper $P_C$ is merged into a list $L = \{P_1, P_2, ..., P_k\}$ with size $K$ containing irrelevant papers. The model is then instructed to summarize ideas from seed $P_B$, search through the papers in $L$, and produce a re-ranked paper list $L'$, following similar criteria described in I3, i.e. novelty, feasibility, and IDR. In I2R, the model is asked to make comparisons to investigate the feasibility of conducting promising IDR when each paper $P_i \in L$ is paired with $P_B$. This task performs better when $P_C$ is at the top of $L'$ after the LLM's re-ranking. Following previous works where ranking is typically performed via pairwise comparison to ensure reliability and robustness (Si et al., 2025), we apply a similar setting using the ranking module shown in the third column in Figure 2.

**Dataset in I2R.** We construct instances in the I2R dataset using a seed paper $P_B$, a list $L$ with $K$ negative papers, and one positive paper $P_C$. The remaining candidates in the list $L$ consist of irrelevant papers randomly sampled from different disciplines.

### 3.3. Data Leakage

We emphasize that the closed-book setting in our benchmark is defined with respect to the target IDR paper $P_A$. The model receives only the titles and abstracts of papers $P_B$ and $P_C$ as input and must decide whether their combination constitutes a valid IDR without access to $P_A$ itself.

Even if $P_B$ and $P_C$ are partially recalled from pretraining, the core task requires the model to perform *input-grounded integration*—reasoning about the synergies between the two given papers—rather than merely retrieving memorized associations. To clarify this perspective, we go further by evaluating the same tasks when prompting the full paper to the LLMs. As described in Appendix A.2, we observe that using the full paper content or only the title and abstract does **not** yield a notable difference in performance. Moreover, we also note that many other works which use papers as a source of information for LLMs have also performed their tasks using only titles and abstracts (Dennstädt et al., 2024; Cohen et al., 2006; Lo et al., 2020).

## 4. Experiment Setup

**Model Setup.** We run the three tasks introduced in Section 3.2 with 10 models, including 5 LLMs with reasoning capabilities. The model temperature is set to 1.0 and maximum output tokens to 1500. We use zero-shot prompting in I2R and apply both zero-shot and few-shot prompting (Brown et al., 2020; Kojima et al., 2022; Wei et al., 2022) in IPI and I3 (with 5 instances in IPI and 3 in I3). We curate few-shot learning examples with a mixture of positive and negative samples from our task datasets. We also ensure that the knowledge cutoff date for the models in our experiments has no overlap with the earliest date of the papers that we collected from ArXiv. Details of the model setup can be found in Appendix A.3.1.

**Task Setup.** In IPI and I3 task, we construct regular expressions to extract LLM answers. For I2R, and following the discussion in (Si et al., 2025), we consider that pairwise comparison can provide better results than asking LLMs directly to predict the final re-ranking decision. We apply the same Swiss tournament used in (Si et al., 2025) with the number of rounds set to 10 and the length $K$ of paper list $L$ to 30. Detailed prompts are provided in Appendix A.9.

**Evaluation Metrics.** For the classification task in IPI and I3, and considering the imbalance nature for our data, we use Macro-F1 score to report the results. For the recommendation task in I2R, we utilize Mean Reciprocal Rank (MRR) (Craswell, 2009). For statistical significance test, we employ the Wilcoxon signed-rank test (Hsieh et al., 2008). We include the details of the evaluation metrics for each task in Appendix A.3.2.

## 5. Evaluation with IDRBench

### 5.1. Results

**IPI Task Performance.** The first two rows of Table 2 report macro-F1 scores on the IPI task referring to zero-shot and five-shot. Compared to the random-guessing baseline, which achieves a macro-F1 score of 0.176, all models ex-

| Model Name | | Non-reasoning Models | | | | | | | Reasoning Models | | | | |
|---|---|---|---|---|---|---|---|---|---|---|---|---|---|
| | | OpenAI | Gemini | Llama 3 | | DeepSeek | Qwen | | OpenAI | | | Claude | DeepSeek |
| | | 4o-mini | 2.0 | 3.1 | 3.3 | v3 | 2.5 | 3 | 5-nano | o3-mini | o4-mini | 4 | r1 |
| Cutoff Date | | Oct 2023 | Oct 2023 | Oct 2023 | Jun 2024 | Jun 2024 | Nov 2024 | Mar 2025 | Dec 2023 | Dec 2023 | Apr 2024 | Jul 2024 | Oct 2023 |
| IPI Macro-F1 | 0-shot | 0.587 | 0.538 | 0.551 | 0.468 | 0.607 | 0.571 | 0.527 | 0.486 | 0.610 | **0.630** | 0.582 | 0.507 |
| | 5-shot | 0.510 | 0.534 | 0.336 | 0.444 | 0.614 | 0.501 | 0.565 | 0.523 | 0.617 | **0.640** | 0.621 | 0.551 |
| I3 Set1 Macro-F1 | 0-shot | 0.750 | 0.666 | 0.696 | **0.814** | 0.769 | 0.793 | 0.782 | 0.254 | 0.372 | 0.495 | 0.563 | 0.526 |
| | 3-shot | 0.765 | 0.434 | 0.519 | 0.822 | 0.635 | **0.860** | 0.746 | 0.302 | 0.411 | 0.492 | 0.694 | 0.450 |
| I3 Set2 Macro-F1 | 0-shot | **0.588** | 0.473 | 0.372 | 0.407 | 0.509 | 0.500 | 0.486 | 0.151 | 0.190 | 0.339 | 0.555 | 0.333 |
| | 3-shot | **0.672** | 0.212 | 0.240 | 0.421 | 0.345 | 0.543 | 0.518 | 0.187 | 0.240 | 0.365 | 0.592 | 0.259 |
| I2R MRR | 0-shot | 0.646* | 0.623* | 0.650* | 0.623* | 0.585 | **0.661** | 0.642* | 0.571 | 0.640* | 0.486 | 0.446 | 0.588 |

*Table 2.* Main results on IDRBench evaluation. For IPI task and two subsets in I3 task, we report results in Macro-F1 scores. For I2R task, we report Mean Reciprocal Rank (MRR) scores. The best results within each column are **bolded** and the best results within a similar group are underlined. Asterisks in the I2R results denote absent of statistical difference to the bolded baseline value.

hibit substantially better performance but **without a strong difference among them**. In the zero-shot setting, we observe macro-F1 scores between 0.486 (`gpt-5-nano`) and 0.630 (`gpt-o3-mini`), with an average of 0.556. Similar variances occurred in the few-shot setting, ranging from 0.510 (`gpt-4o-mini`) to 0.640 (`gpt-o4-mini`), with an average of 0.483. Interestingly, few-shot augmentation does not consistently lead to performance improvements and may even degrade performance in some cases, as observed for `llama-3.1-70B-instruct` and `gpt-4o-mini`.

**I3 Task Performance.** For I3, we report Macro-F1 results regarding two subsets constructed based on different negative sampling strategies. For I3-Subset 1 that focuses on IDR feasibility evaluation (negative samples are from different disciplines), `llama-3.3-70B-instruct` and `Qwen-2.5-Instruct` achieve the highest score among the original 10 models tested, achieving 0.814 and 0.860 on zero-shot and few-shot respectively. On I3-Subset 2 (negatives from the same discipline), all models yielded poor results compared to I3-Subset 1, which showcases that the models on average are less capable of recognizing that two papers from the same discipline do not form a valid interdisciplinary research, despite their feasibility to be integrated in a single discipline context.

**I2R Task Performance.** The final row in Table 2 presents I2R recommendation results in IDRBench. We apply the Mean Reciprocal Rank (MRR) metric, which emphasizes ranking accuracy by rewarding correct items for being ranked higher. To ensure the reliability of our results, we perform the statistical significance tests when comparing

to the best results (highlighted in bold) using the Wilcoxon signed-rank test (Hsieh et al., 2008) – marked with asterisks in the table. All models reach similar results, i.e., approximately a 0.670 MRR.

### 5.2. Error Analysis on I3 Task

To provide a more granular analysis of the IDR integration mechanism (Task I3), we partition the ground truth labels, calculating the True Positive Rate (TPR) for positive instances and the True Negative Rate (TNR) to negatives for subset-1 and subset-2, showing the decomposed performance in Figure 4.

Considering the current release models, we observe that some are more optimistic toward IDR integration, producing positive predictions more frequently (higher TPR) while also producing more mistakes on negative instances (lower TNR). For instance, in Figure 4-b, `gemini-2.0-flash` seems very optimistic, reaching almost 100% TPR and around 42% TNR, but it achieves only 15% TNR for Subset2. This behavior is also observed for `gpt-5-Nano` and `o3-mini` in the zero-shot setting, and for `gpt-5-Nano` and `o3-mini` in the few-shot setting. In contrast, `claude-sonnet-4` and `gpt-o4-mini` appear more pessimistic (in topic integration), exhibiting a low TPR and a high TNR. This tendency toward optimism or pessimism appears more pronounced in reasoning models, across both zero-shot and few-shot settings.

To propose a potential explanation for this optimistic and pessimistic behavior, we follow the findings from recent

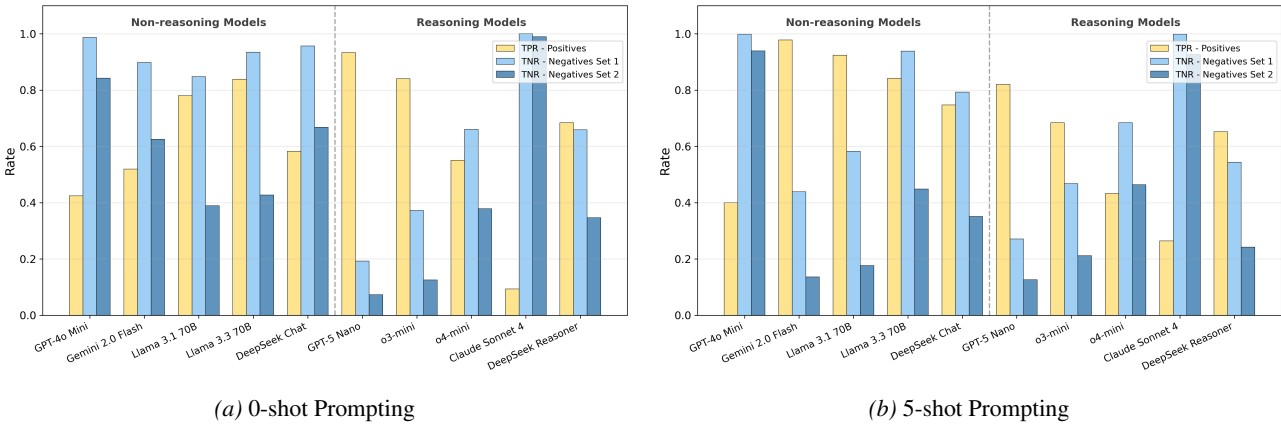

*(a)* 0-shot Prompting               *(b)* 5-shot Prompting

*Figure 4.* Comparison of True Positive Rate (TPR) and True Negative Rate (TNR) for Task I3 across different models.

studies (West & Potts, 2025; Murthy et al., 2025) that alignment phase in post-training models has a negative effect on creative tasks. Thus, our analysis in I3 suggests an association between error types and different alignment targets: on the spectrum of alignment targets between usability and safety, models that are post-trained on usability (OpenAI, 2024; 2025; Hassabis & Kavukcuoglu, 2024) (including `o3-mini`, `o4-mini`, `gpt-5-nano`, `gemini-2.0-flash`, and `llama-3-instruct` models) tend to exhibit more optimistic behavior, thus yielding more false positive answers. On the other hand, when the post-training target leans toward guardrailing safety and integrity (OpenAI et al., 2024; Anthropic, 2025), models like `claude-4` and `gpt-4o-mini` tend to yield more conservative responses, yielding more negative responses. For models that balance between those two objectives, such as `deepseek-r1` and `deepseek-v3`, they have a balanced attitude and awareness in IDR tasks.

Considering the reasoning models performance, we observe a lower precision on two subsets of I3 than their non-reasoning counterparts. In Table 2, I3 Set1, non-reasoning LLMs exhibit a wider performance margin in IDR integration, with scores around 0.6 and 0.8, while reasoning ones only score around 0.2 and 0.6. In Figure 4 we also observe a less sensitive behavior in TPR/TNR shift when provided with few-shot contexts. Such degraded performance on reasoning models echoes findings in (Luyten & van der Schaar, 2026; Mohammadi, 2024), where they present a reasoning-creativity trade off for those reasoning-based LLMs.

Our observations further support the opinion that a model's reasoning efforts often act as constraints and blurs the actual input context when facing creative tasks (I3 in our case), which potentially undermines the models' performance.

### 5.3. Human Evaluation

We gather a group of 6 human experts from diverse academic backgrounds to assess the LLM generated IDR idea quality that consists of 60 idea samples. This evaluation aims to understand the usefulness of LLM generated ideas when assisting human researchers to promote the 'Eureka' moment in idea integration. The IDR ideas are generated in the form of a running abstract and a sentence that describes the key integration logic, as shown in the I3 output in Figure 2. Following Figure 4-b, we use `gemini-2.0-flash`, as it has a more optimistic behavior in I3. The idea quality is assessed from the following two dimensions: the correctness of the idea (concerning the integration of papers $P_B$ and $P_C$) and the clarity of the idea descriptions (concerning the facility to understand how the areas are integrated). Each of these dimensions is evaluated on a scale of 1 (lowest score) to 5 (highest score), adapted from the review process in (Si et al., 2025). The annotators are also asked to provide a confidence rating for each rated abstract and integration sentence.

Figure 5 showcases the average rating from human evaluation results on 60 idea samples, including the standard error statistics. For both LLM generated IDR abstract and integration sentences, the experts provide positive ratings, indicating that the idea samples are useful in assisting human researchers. Details of the human evaluation results can be found in Appendix A.5.

### 6. Conclusion

We introduce IDRBench, a novel benchmark to probe LLMs' capacity for Interdisciplinary Research (IDR). IDRBench includes a high-quality dataset collected from human annotation. We also design a set of three progressive tasks: *(i) IDR Paper Identification (IPI)*, *(ii) IDR Idea Integration (I3)*, and *(iii) IDR Idea Recommendation (I2R)*. Throughout the experiment analysis, we uncover that LLMs are capable of

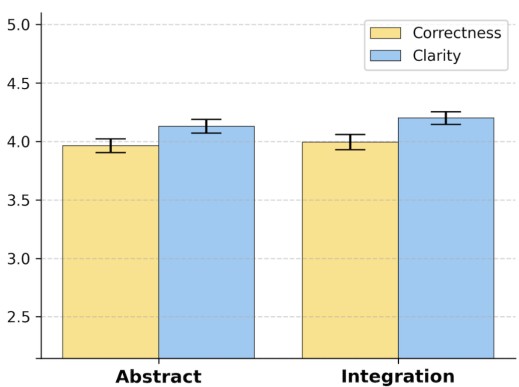

*Figure 5.* Human evaluation of LLM-generated IDR ideas for correctness and clarity. Scores (1-5) were assigned to abstracts and integration sentences. We use `gemini-2.0-flash`, the most optimistic model in I3 results to generate both abstract and integration sentence and evaluate on human experts.

integrating ideas as well as generating correct and clear IDR descriptions, but they still lack certain awareness of ideas that are feasible but not IDR. We further demonstrate that the models undergo a shift from over-estimation to a conservative behavior in I3; we also show that reasoning-oriented post-training introduces a systematic bias when performing integration. To the best of our knowledge, IDRBench is the first to rigorously investigate LLMs' IDR ability and to provide insights that encourage follow-up studies. Moreover, future frameworks may leverage IDRBench by: (i) utilizing the dataset to evaluate models with specific temporal cutoffs, thereby preventing data contamination; (ii) exploring the benchmark's proposed tasks; and (iii) incorporating our analytical insights regarding LLM behavior into their framework design.

## 7. Limitations

While ArXiv provides a reliable foundation for evaluating IDR with LLMs, expanding to a broader range of disciplines would further fortify the relevance and generalizability of our findings. We also acknowledge that the I3 and I2R tasks inherently involve a large and subjective solution space—a characteristic shared by many well-established NLP benchmarks such as text summarization, open-ended generation, and dialogue systems. Our benchmark adopts a similar philosophy by: (i) grounding the evaluation in real, traceable IDR papers authored by human researchers, (ii) employing expert-validated positive instances with documented integration logic, and (iii) complementing automated metrics with multi-dimensional human evaluations. We believe that establishing IDR benchmarks is necessary even when the solution space is broad, and IDRBench represents a careful first step toward a rigorous and controllable entry point for

evaluating LLMs within the open-ended domain of interdisciplinary research.

## Acknowledgements

This work was funded by the Artificial Intelligence for Design Program (AI4D-156) of the National Research Council Canada (NRC). The second and third authors' work was supported by Brazilian funding from IFG, CNPq (443011/2023-0), and INCT-TILD-IAR (408490/2024-1).

## Impact Statement

While IDRBench aims to advance the understanding of LLMs' capabilities in interdisciplinary research, the technology it evaluates carries potential risks. LLM-generated research ideas could be misused to produce misleading or superficially plausible but scientifically unsound research proposals. The ability to automate interdisciplinary ideation could also be exploited for academic misconduct, such as generating fraudulent grant proposals or papers. Furthermore, over-reliance on LLM-generated ideas without adequate expert oversight could propagate biases present in training data across disciplines. We encourage future work to address these risks through robust verification protocols and responsible deployment guidelines.

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

# A. Appendix

## A.1. Dataset Details

This section provides details regarding the IDRBench dataset, our custom web platform, and the background of our annotators.

### A.1.1. ANNOTATION PLATFORM

To collect the positive samples in IDRBench, we designed a custom Web platform that includes four tasks for the annotators to complete. Specifically, the annotators are first provided with a panel of papers shown in Figure 6 when logged in, where they can choose the paper that they feel comfortable annotating. In task one, shown in Figure 7, they are instructed to first read the title and abstract of a given paper and decide whether it is an IDR paper. If so, they are directed to task two (Figure 8) to indicate the research type (i.e., whether it is basic research or applied research) of the IDR paper. In the third task, the annotators are asked to specify the exact papers that contribute to the IDR idea in this paper. They can add up to 4 papers that describe the IDR idea, as shown in the screenshot of Figure 9. Finally, they are asked to annotate the specific sentence(s) in this IDR paper that specifically describe such integration, as shown in Figure 10. The Web platform is also designed in a way where the annotators can revert any progress they have made so far in case they change their minds, shown in Figure 11.

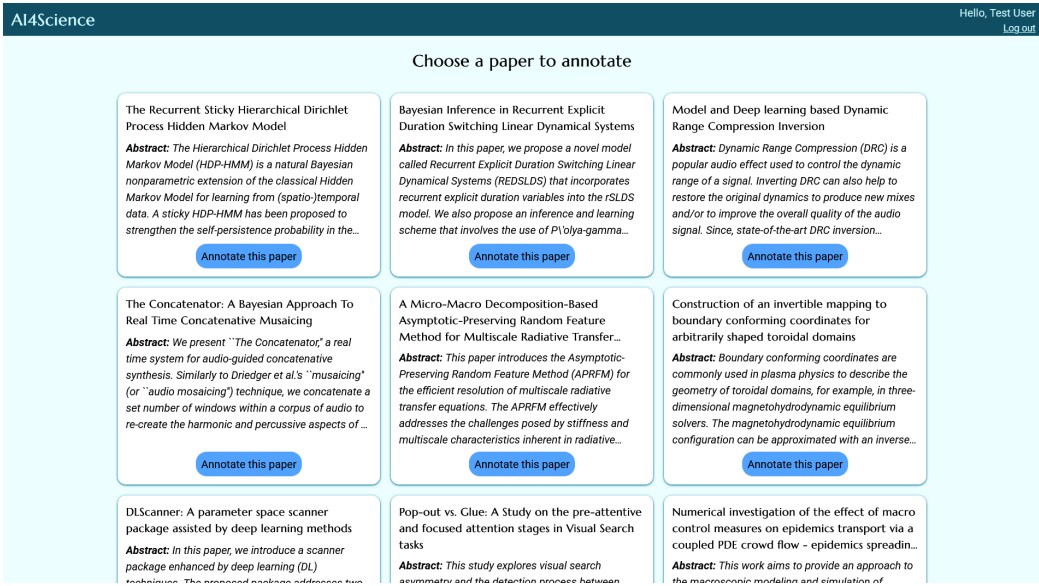

*Figure 6.* List of papers available for the annotator to choose from.

### A.1.2. HUMAN ANNOTATION

We recruit 9 students from diverse academic backgrounds to perform the human annotation. Each student received about $18 for each work hour. The academic level of the annotators ranges from the fourth year of undergraduate studies to the second year of PhD. Table 3 summarizes the annotators' background, including their level of study and area of expertise. To ensure mutual agreement among annotators, each paper triplet sample is annotated twice, and we only keep triplets that is marked positive by two annotators.

| Level of study | Count | Expertise Area |
|---|---|---|
| Undergraduate | 3 | Robotics, Control, AI, Natural Language Processing, |
| Master's | 2 | Robotics, Rehabilitation, AI, Medical science, |
| | | Mechanical and Materials Engineering, Mechatronics and Robotics Engineering, |
| PhD | 4 | Brain-computer interfaces, Machine Learning |

*Table 3.* Academic background summary of annotators.

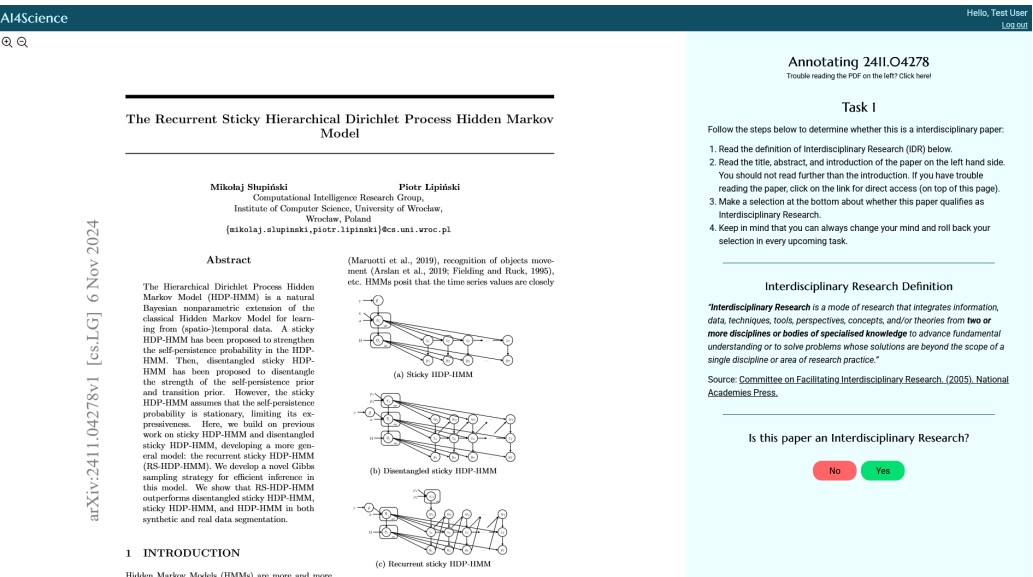

*Figure 7.* Task 1, displayed after the annotator selects a paper.

## A.2. Data Leakage on I2R tasks

In order to empirically address the issue of data leakage in Section 3.3, we conducted a ablation study to test model's performance using a subset of I3 task (200 random samples) in IDRBench. We use the zero shot prompting setup for our ablation study. Table 4 showcase the results of our ablation study. We observe that the results are often consistent across all five models tested. The outcome further supports our argument that the body of those papers are already in (or highly overlapped with) LLM pre-trained memory.

| | gpt 4o-mini | gpt o3-mini | gemini 2.0 flash | llama-3.1-70b-instruct | llama-3.3-70b-instruct |
|---|---|---|---|---|---|
| Title + Abstract | 0.734 | 0.683 | 0.782 | 0.780 | **0.795** |
| Full Content | 0.723 | 0.699 | 0.732 | **0.739** | 0.723 |

*Table 4.* Ablation study on I3 with different input context. Performance reported in F1 Score.

## A.3. Experiment Details

This section introduces the details of our experiment setup using IDRBench. Section A.3.1 describes the models that we use in our evaluation, Section A.3.2 lists the evaluation metrics used to quantify the performance of different models, and Section A.9 provides the prompts that we used in our experiments.

### A.3.1. MODELS

**ChatGPT.** First released in November 2022, `ChatGPT` is a series of models that is trained by OpenAI to conduct complex tasks using natural language prompts (OpenAI et al., 2024; OpenAI, 2024). In our paper, we select two types of GPT models from the GPT family, where `gpt-4o-mini` focuses on instruction following abilities, and `gpt-o1-mini` and `gpt-o3-mini` have enhanced reasoning capabilities in complex tasks.

**Claude-Sonnet.** This is the latest model released by Anthropic that incorporates an adaptive thinking process when prompted with a task. Specifically, it simulates an adaptive switch between the system 1 thinking and system 2 thinking, akin to human (Anthropic, 2025). We use the latest model `claude-sonnet-4-20250514` to run our benchmark.

**Llama-3-Instruct.** As an updated version from `llama-2` (Grattafiori et al., 2024), `llama-3` is trained with more recent corpora from various sources and achieves a better performance in various benchmarks. Different from the GPT family,

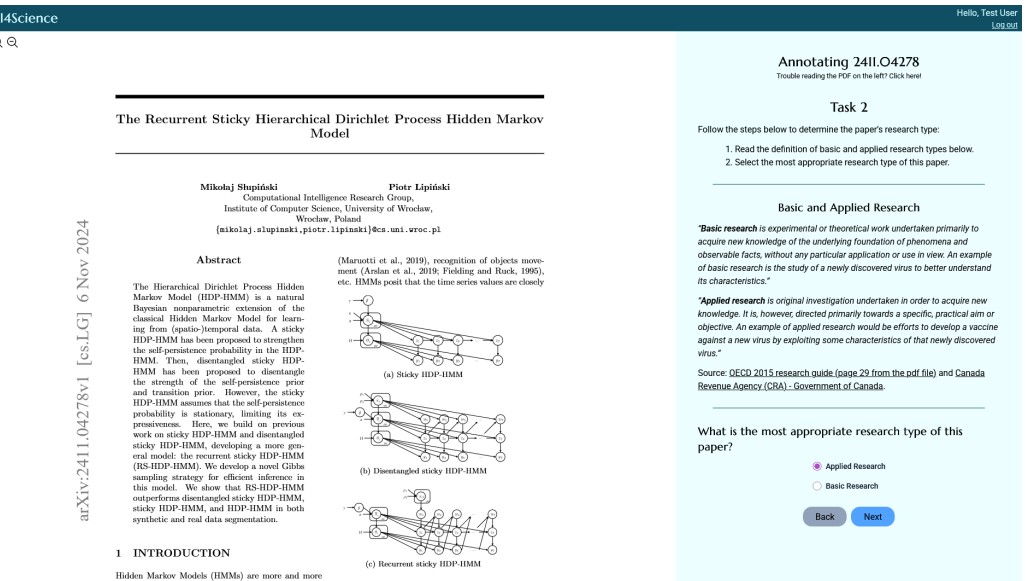

*Figure 8.* Task 2, displayed if the annotator answers "Yes" in Task 1.

Llama models are completely open-source. In our evaluation, we use `llama-3.1-70B-instruct` and `llama-3.3-70B-instruct`, the two most powerful models in the family so far.

**Deepseek.** Deepseek is a suite of models trained under a schema that has higher efficiency yet with a comparable performance to those closed-source models (DeepSeek-AI et al., 2025). The Deepseek family includes a basic chat model called `deepseek-v3` and an advanced reasoning model `deepseek-r1`. We include both the chat model and the reasoning model in the results.

| Model | Cutoff Date | Release Date |
|---|---|---|
| `gpt-4o-mini` | Oct 2023 | Jul 2024 |
| `gpt-o3-mini` | Oct 2023 | Jul 2024 |
| `gpt-o4-nano` | Oct 2023 | Aug 2025 |
| `gpt-5-nano` | Oct 2023 | Aug 2025 |
| `llama-3.1-70B-Instruct` | Dec 2023 | Jul 2024 |
| `llama-3.3-70B-Instruct` | Dec 2023 | Jul 2024 |
| `gemini-2.0-flash` | Jun 2024 | Dec 2024 |
| `claude-4-Sonnet` | Nov 2024 | Feb 2025 |
| `deepSeek-v3` | Jul 2024 | Dec 2024 |
| `deepSeek-r1` | Oct 2023 | Dec 2024 |
| **Earliest paper in IDRBench** | Nov 2024 | |

*Table 5.* Knowledge cutoff dates of LLMs' pretraining data.

### A.3.2. EVALUATION METRICS

We use a set of mainstream metrics for both our classification tasks and recommendation tasks. For the classification tasks, we report F1 and Macro F1 scores, and for the recommendation tasks, we report MRR. MRR measures, on average, how far down the ranked list the first relevant item appears. Specifically, for a set of queries $Q$, let $\text{rank}_q$ be the position of the first relevant item for query $q \in Q$. The Reciprocal Rank (RR) for $q$ is

$$\text{RR}_q = \begin{cases} \dfrac{1}{\text{rank}_q}, & \text{if a relevant item is found,} \\ 0, & \text{otherwise.} \end{cases}$$

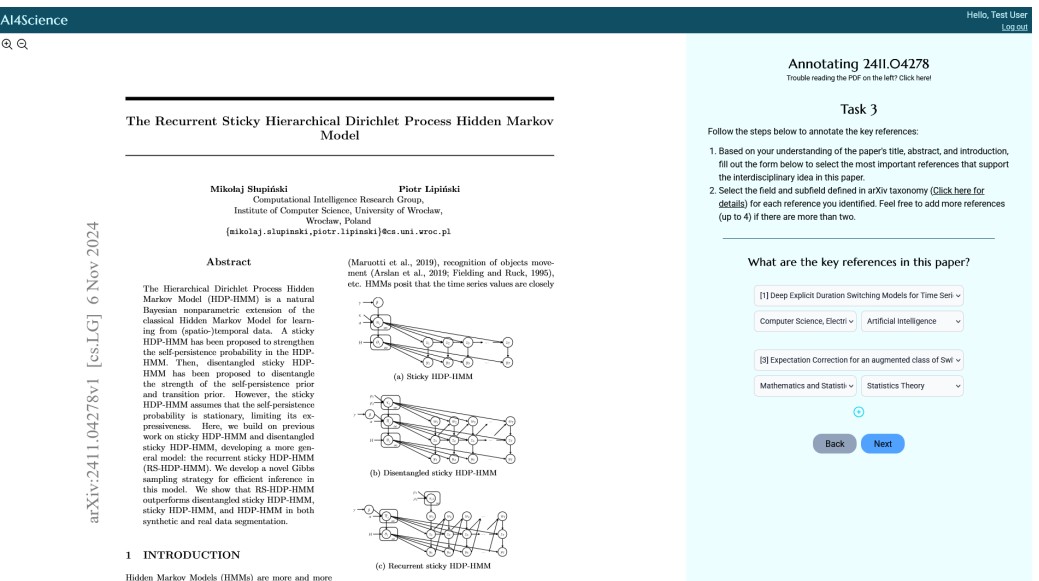

*Figure 9.* Task 3, displayed after completing Task 2.

The Mean Reciprocal Rank is then

$$\text{MRR} = \frac{1}{|Q|} \sum_{q \in Q} \text{RR}_q = \frac{1}{|Q|} \sum_{q \in Q} \frac{1}{\text{rank}_q}.$$

### A.4. Error Analysis

In this section, we provide detailed analysis for LLM's divergent attitude (i.e. models being optimistic to provide more false positive answers v.s. models being pessimistic to provide more false negative answers) observed in our decomposed performance analysis in I3 task.

Table 6 showcase the comparison chart on the post-training strategies summarized from the technical reports of the models evaluated in our experiments. In the row titled 'Refusal Style', it demonstrates a clear distinction of model's tendency when providing answers to various tasks. When we sort the models according to their refusal style on the spectrum that stretches from being **safe** (i.e. providing responses with high refusal) to being **helpful** (i.e. answer as much as the model can), we observe a high alignment with our observed performance breakdown in I3: when models showcase a low True Positive Rate (TPR) in integration decisions (reported in Figure 4), such as `gpt-4o-mini` and `claude-sonnet-4`, they all falls in the more conservative side of the spectrum. In contrast, when models showcase high TPR, their refusal style tends to fall in the more helpful side of the spectrum, including `llama-3-instruct` models, `gemini-2.0-flash`, `o3-mini`, `o4-mini`, and `gpt-5-nano`. These models are explicitly post-trained on either being helpful or providing useful answers to user's prompts. Finally, when we look at models that employs a relatively balanced post-training strategy such as `deepseek-r1` and `deepseek-v3`, we see their TPR also ranks in the middle among all the tested models.

In addition to the observation regarding the different post-training strategies of those models, it is also noted that such observation maintains consistent under both zero-shot prompting as well as few-shot prompting schema, indicating that adding In-Context Learning (ICL) methods has little influence on response outcome of I3 tasks. We hereby leave the quantitative study of this matter as a separate line of research in our future work.

### A.5. Human Evaluation

This section introduces the details of the human evaluation on LLM generated idea quality mentioned in Section 5.3. To ensure the reliability of evaluation outcomes, we only include experts that have at least post-undergraduate education background in Table 3, which accounts for a total of 6 experts. We pick `gemini-2.0-flash` to generate the running abstract as well as the integration sentence due to its outstanding performance in achieving the highest TPR in I3 task performance breakdown, which is shown in Figure 4. The reason we focus on TPR specifically is due to the data imbalance

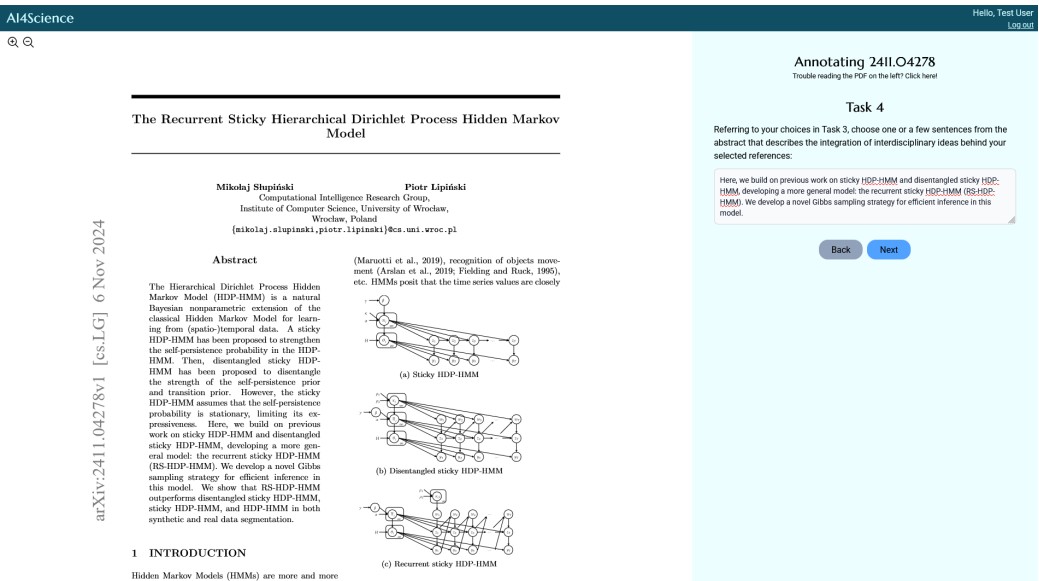

*Figure 10.* Task 4, displayed after completing Task 3.

| Feature | GPT-4 (OpenAI) | GPT-o series | Llama 3 (Meta) | Claude 3/4 (Anthropic) | Gemini 2 (Google) | DeepSeek-V3 (DeepSeek) |
|---|---|---|---|---|---|---|
| **Primary Lean** | Safety & Reliability | Usability & Helpfulness | Usability & Helpfulness | Integrity & Agency | Agentic Utility & Reasoning | Technical Reasoning & Efficiency |
| **Alignment Method** | RLHF + RBRMs | RLHF + RBRMs | RLHF + DPO | Constitutional AI (CAI) | RL-based Thinking | GRPO (Group RL) |
| **Refusal Style** | Conservative (High) | Conservative (Low) | Nuanced (Aims to answer) | Philosophical (Contextual) | Non-intrusive | Pragmatic (Low refusal) |
| **Technical Focus** | Human-in-the-loop safety | Human-in-the-loop safety | Addressing over-refusal | Reducing "alignment faking" | Multi-step tool use | Math, Code, & MoE Balancing |

*Table 6.* Comparison of the post-training target of tested LLMs with their technical focus.

nature in I3, where we maintain a positive to negative ratio of 1:10. The prompt that we used to generate the idea can be found in Table 19.

### A.5.1. IDR IDEA QUALITY RATING

To obtain the rating on IDR idea quality, we provide the instructions in Table 7 and the evaluation form in Figure 12 to obtain the ratings. Table 8 showcases the numeric results from human evaluation. The average correctness and clarity rating of the running abstract reaches 3.963 and 4.129 with relatively small standard errors, meaning that the annotators have a high agreement on the utility of the generated ideas. Also, on a scale of 1 to 5, the rating falls in the top 25 percent quartile, indicating the ideas are being helpful in integrations that lead to promising IDR ideas. Another observation is that human experts tend to find the integration sentence of higher quality than the running abstract, with the average consensus reach 3.993 and 4.200 in terms of correctness and clarity. Finally, the annotators exhibit a relatively high average confidence towards their ratings. This is justified from the STEM background of them and a high alignment between their expertise and the IDR idea topics, given our papers are derived from the Arxiv platform.

### A.5.2. USER-CENTRIC STUDY WITH DOMAIN RESEARCHERS

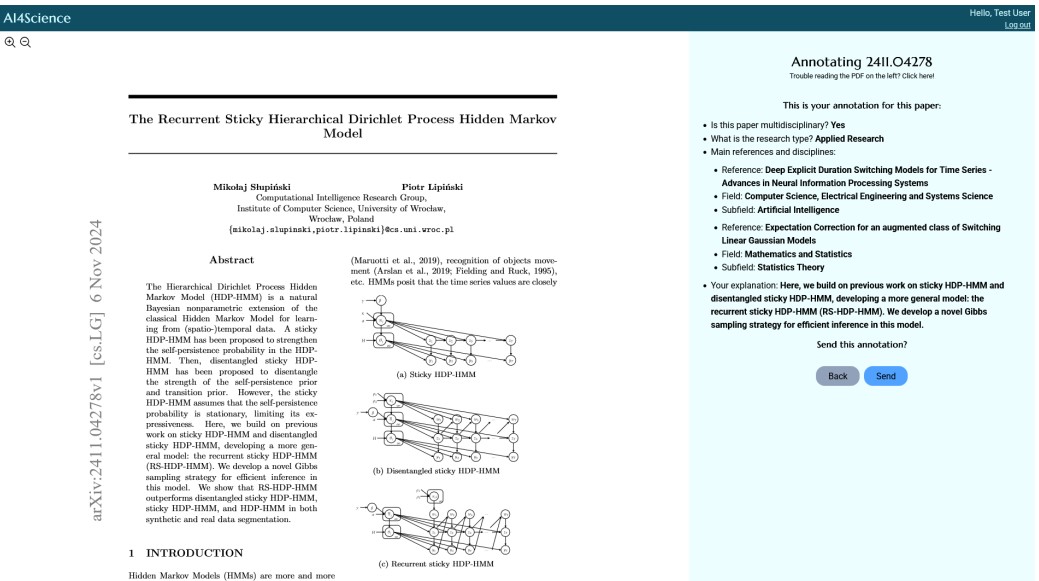

*Figure 11.* Review page, where the annotator can review and optionally go back and edit their answers.

---

### Human Evaluation Instructions

\* Read the title and abstract of papers from **Discipline A** and **Discipline B** to get a brief sense of the research background. (You **DO NOT NEED to know** the exact details of these abstracts).
\* Read the abstract and integration explanation generated by the model and rate its quality from the two dimensions above.
\* Each of the two evaluation dimensions has detailed explanations attached in the question. Please read carefully.
\* You only need to select the rating, **DO NOT FILL** in the text box.

---

*Table 7.* Instructions provided to human experts on IDR idea evaluation.

To further validate the practical utility of our benchmark in real-world research scenarios, we conducted an additional user study involving 56 researchers across nine major disciplines at an academic institution. We applied IDRBench tasks to the researchers' own published work: I2R was used to identify paper pairs with interdisciplinary collaboration potential, and I3 was then applied to generate research proposals based on those pairs. The generated proposals were sent directly to the original authors for evaluation across three criteria—Integration, Clarity, and Interest—on a 1–5 scale. We collected 80 proposal evaluations. Score distributions (1–2, 3, 4–5) were as follows: Integration (34, 15, 31); Clarity (24, 16, 40); and Interest (37, 13, 30). The Clarity criterion achieved the highest positive concentration, with over 50% of ratings at scores 4–5, confirming that LLM-generated IDR proposals are generally well-formulated and understandable. Integration and Interest scores exhibited a bimodal distribution concentrated at the extremes, reflecting the inherently subjective nature of interdisciplinary synergy assessment. Consistent with the results in Figure 5, this user study confirms that a substantial portion of LLM-generated ideas are perceived as valuable by domain experts, while also underscoring the variability inherent in evaluating creative, cross-disciplinary proposals.

### A.6. I2R Kendall Correlation Analysis

To verify that LLM re-ranking decisions in the I2R task are not primarily driven by surface-level semantic similarity between the seed paper $P_B$ and each candidate paper, we compute Kendall's $\tau$ correlation between each model's re-ranking and a ranking based on SciBERT semantic similarity scores. Table 9 reports the results. Correlations range from approximately 0.18 to 0.25 across all models, indicating a weak relationship at best. This confirms that LLMs are not simply ranking candidate papers by topical similarity to the seed paper; instead, they rely on a deeper assessment of interdisciplinary integration feasibility.

*Figure 12.* Screenshot of the questions in human evaluation on IDR quality. For each abstract and integration sentence, annotators will be asked the same group of questions.

|  | **Correctness** | **Clarity** | **Confidence** |
|---|---|---|---|
| Running Abstract | 3.963 ($\pm$) 0.058 | 4.129 ($\pm$) 0.059 | 4.159 ($\pm$) 0.032 |
| Integration Sentence | 3.993 ($\pm$) 0.064 | 4.200 ($\pm$) 0.054 | 4.339 ($\pm$) 0.037 |

*Table 8.* Average rating consensus with standard error (numbers after $\pm$) on IDR idea quality. We use `gemini-2.0-flash`, the most optimistic model in I3 results to generate both abstract and integration sentence and evaluate on human experts. We also include the average confidence with standard error from human experts after they report their ratings.

### A.7. Model Output Comparison

In addition to the reporting of similarity scores in Section 5, we also conduct a qualitative analysis on the LLM output reasons against human researchers' original writing on formulating a potential IDR research. Our analysis is twofold: on one hand, we compare the key integration reasoning from LLM output with the annotated sentences marked by the annotators; on the other hand, we compare the full abstract generated by LLM with the abstract of the targeted IDR paper. Figure 14 provides a comparison between the annotation and the LLM-generated response.

| Model | Kendall's $\tau$ |
|---|---|
| `gpt-4o-mini` | 0.196 |
| `gpt-o3-mini` | 0.180 |
| `gemini-2.0-flash` | 0.220 |
| `llama-3.3-70B-instruct` | 0.211 |
| `claude-4-sonnet` | 0.249 |
| `deepseek-r1` | 0.194 |

*Table 9.* Kendall's $\tau$ correlation between I2R model re-rankings and rankings based on SciBERT similarity scores. Low correlations confirm that surface-level paper similarity does not dominate LLM re-ranking decisions.

## A.8. Scalability of IDRBench

To further explore the potential of our IDRBench dataset, including its scalability and versatility on model fine-tuning, we take the annotated positive samples from I3 and derive a training dataset, which is thereafter utilized to fine-tune a `llama-3.1-8B-instruct` model. We put the details of the data augmentation as well as the model fine-tuning implementations in the sections below.

### A.8.1. DATA AUGMENTATION

We first take the positive pairs from the paper triplet $[P_A; (P_B, P_C)]$ described in Figure 1 and only augment the $(P_B, P_C)$ section of the triplet. Specifically, we use `gpt-5.2-latest` as the teacher model to rewrite the abstract of either $P_B$ or $P_C$. We ask the model to provide two different rewrite versions of the abstract for $P_B$ and $P_C$ respectively. Table 20 showcases the abstract rewrite prompt of the positive samples. We then pair the rewritten abstract with each other, meaning that for each positive paper pair $(P_B, P_C)$, there will be 8 more positive pairs augmented into the dataset.

For the negative samples within the I3-Subset 2, we curate another prompt using the same LLM to augment the reason of why the pair fails to derive an IDR idea. Table 21 list the detailed prompt. To ensure the model learns sufficient structures of instruction following, we further augmented an additional 1,000 data samples from SlimOrca (Mukherjee et al., 2023), making the total size of the dataset reaches 3,473 samples.

### A.8.2. MODEL FINE-TUNING

We then use the dataset to fine-tune a local `llama-3.1-7B-instruct` model using LoRA (Hu et al., 2022), and we include the training hyperparameters and the plot of training loss in Figure 13. From the figure, we can see `llama-3.1-7B-instruct` converges quickly in the first 100 steps and then remain stable until termination. We then use this fine-tuned model and tested on the full IDRBench IPI task as well as the held-out dataset in I3. We report the results in Table 10.

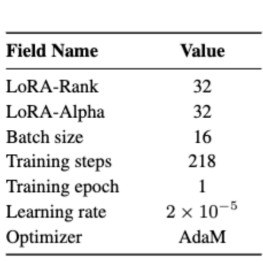

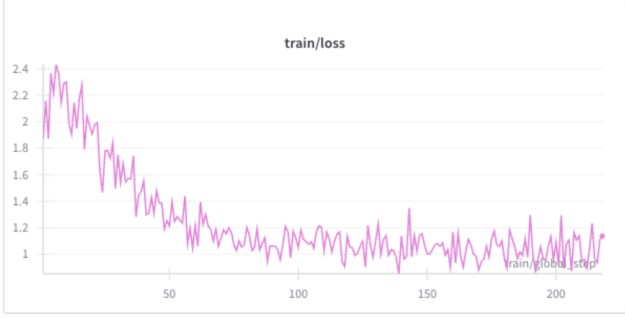

| Field Name | Value |
|---|---|
| LoRA-Rank | 32 |
| LoRA-Alpha | 32 |
| Batch size | 16 |
| Training steps | 218 |
| Training epoch | 1 |
| Learning rate | $2 \times 10^{-5}$ |
| Optimizer | AdaM |

*Figure 13.* Table of fine-tuning hyperparameters (on the left) and training loss plot (on the right).

Based on the results, we can see that `llama-3.1-8b-instruct` showcase a performance increase on both subset 1 and subset 2 evaluations on I3 task, achieving an average of 12.72% increase in Marco-F1 scores. However, the performance is relatively on par when evaluated on IPI task. The performance results on `llama-3.1-8b-instruct` indicate that

the augmented IDRBench dataset has the potential to improve model performance on those IDR tasks, making our dataset versatile and scalable for future development of more powerful LLMs that excel in the IDR context.

|  | IPI | I3-Subset 1 | I3-Subset 2 |
|---|---|---|---|
| Baseline | **0.094** | 0.664 | 0.539 |
| Fine-tuned (LoRA) | 0.089 | **0.796** | **0.569** |

*Table 10.* Fine-tuning results of `llama-3.1-8b-instruct` on three IDRBench tasks. Results are reported in Marcro-F1 scores.

## A.9. Prompts

In this section, we list the prompts that we used in our model evaluation, including 1) IDR Paper Identification (IPI), 2) IDR Idea Integration (I3), and 3) IDR Idea Recommendation (I2R). We also include prompts that are used to generate running abstract and integration logic, which is thereafter used in human evaluation.

---

**IPI prompt**

---

Read the title and abstract of a given academic paper and identify whether this is an interdisciplinary research paper. Also, select one or more subjects from the list below to indicate which subject(s) does this paper belong to. After you provide your verdict and your choice, provide a score from 0 to 100 to indicate your confidence level in the correctness of the verdict.
The official definition of a typical interdisciplinary paper can be found below:
Interdisciplinary Research is a mode of research that integrates information, data, techniques, tools, perspectives, concepts, and/or theories from two or more disciplines or bodies of specialised knowledge to advance fundamental understanding or to solve problems whose solutions are beyond the scope of a single discipline or area of research practice.
Think carefully to make your verdict, answer "Yes" when this is a valid IDR paper. Otherwise, answer "No".
Note: The confidence level indicates the degree of certainty you have about your verdict and is represented as a percentage. For instance, if your confidence level is 80, it means you are 80 percent certain that your answer is correct and there is a 20 percent chance that it may be incorrect.

Paper title: %s;
Paper abstract: %s;

Subject list: ["Computer Science, Electrical Engineering and System Science", "Economics and Quantitative Finance", "Mathematics and Statistics", "Physics", "Quantitative Biology", "Other"]

Use the template (in this format, with no markdown and lines separated by '\n') below to provide your answer.
Your verdict: A simple answer containing either "Yes" or "No".
Confidence score: A numeric score ranging from 0 to 100
Subject: Your choice of subjects from the list above. Use a list with square brackets "[]" separated by comma and remember to use "" to wrap your answer.

---

*Table 11.* Prompt used for the IPI task.

**I3 prompt**

Read the title and abstract of papers from two disciplines and decide whether you can extract concepts from both disciplines to create a novel multidisciplinary research idea. After you provide your verdict, provide a score from 0 to 100 to indicate your confidence level in the correctness of the verdict.

Keep in mind a good Interdisciplinary Research idea includes the following standards:

* This research idea should be Interdisciplinary, whereas the idea stems from the combination of ideas from the two papers introduced above.

* The Interdisciplinary Research ideas should follow this definition: Interdisciplinary Research is a mode of research that integrates information, data, techniques, tools, perspectives, concepts, and/or theories from two or more disciplines or bodies of specialised knowledge to advance fundamental understanding or to solve problems whose solutions are beyond the scope of a single discipline or area of research practice.

* This research idea should be feasible, whereas the hypothesis is not purely theoretical and can be validated by experiments.

* This research idea should be novel, whereas it is not only rare but also ingenious, imaginative, or surprising.

* This research idea should be useful, whereas it applies to the stated problem and is effective at solving the problem.

Think carefully to make your decision, and you should only answer "Yes" when this multidisciplinary idea meets ALL of the standards above. Otherwise, you should answer "No".

Note: The confidence level indicates the degree of certainty you have about your verdict and is represented as a percentage. For instance, if your confidence level is 80, it means you are 80 percent certain that your answer is correct and there is a 20 percent chance that it may be incorrect.

Paper in Discipline 1:
%s

Paper in Discipline 2:
%s

Use the template (in this format, with no markdown and lines separated by '\n') to provide your answer.
Your verdict: A simple answer containing either "Yes" or "No".
Your reason: A short paragraph less than 50 words briefly describes your reasons that you made the verdict above.
Confidence score: A numeric score ranging from 0 to 100

*Table 12.* Prompt used for the I3 task.

---

**I2R prompt (0-shot)**

---

In this task, you are given a main paper introducing the key concepts that provides certain parts in a Interdisciplinary idea as well as two candidate papers that forms the remaining parts of a Interdisciplinary idea. Compare them and select which one is better to pair with the main paper in forming a multidisciplinary idea. After you provide your selection, provide a score from 0 to 100 to indicate your confidence level in the correctness of making this choice.

Keep in mind a good Interdisciplinary Research idea includes the following standards:

* This research idea should be Interdisciplinary, whereas the idea stems from the combination of ideas from the two papers introduced above.

* The Interdisciplinary Research ideas should follow this definition: Interdisciplinary Research is a mode of research that integrates information, data, techniques, tools, perspectives, concepts, and/or theories from two or more disciplines or bodies of specialised knowledge to advance fundamental understanding or to solve problems whose solutions are beyond the scope of a single discipline or area of research practice.

* This research idea should be feasible, whereas the hypothesis is not purely theoretical and can be validated by experiments.

* This research idea should be novel, whereas it is not only rare but also ingenious, imaginative, or surprising.

* This research idea should be useful, whereas it applies to the stated problem and is effective at solving the problem.

Note: The confidence level indicates the degree of certainty you have about your verdict and is represented as a percentage. For instance, if your confidence level is 80, it means you are 80 percent certain that your answer is correct and there is a 20 percent chance that it may be incorrect.

Main paper title: %s;
Main paper abstract: %s;

Paper 1 title: %s;
Paper 1 abstract: %s;

Paper 2 title: %s;
Paper 2 abstract: %s;

Use the template (in this format, with no markdown and lines separated by '\n') to provide your answer.
Your choice: A simple answer containing either "Paper 1" or "Paper 2".
Confidence score: A numeric score ranging from 0 to 100

---

*Table 13.* Prompt used for the I2R task.

## IPI Few-shot Examples

Example 1:
Paper title: Designing a Light-based Communication System with a Biomolecular Receiver; Paper abstract: Biological systems transduce signals from their surroundings in numerous ways. This paper introduces a communication system using the light-gated ion channel Channelrhodopsin-2 (ChR2), which causes an ion current to flow in response to light. Our design includes a ChR2-based receiver along with encoding, modulation techniques and detection. Analyzing the resulting communication system, we discuss the effect of different parameters on the performance of the system. Finally, we discuss its potential design in the context of bio-engineering and light-based communication and show that the data rate scales up with the number of receptors, indicating that high-speed communication may be possible.
Your verdict: Yes

Example 2:
Paper title: BarcodeMamba: State Space Models for Biodiversity Analysis; Paper abstract: DNA barcodes are crucial in biodiversity analysis for building automatic identification systems that recognize known species and discover unseen species. Unlike human genome modeling, barcode-based invertebrate identification poses challenges in the vast diversity of species and taxonomic complexity. Among Transformer-based foundation models, BarcodeBERT excelled in species-level identification of invertebrates, highlighting the effectiveness of self-supervised pretraining on barcode-specific datasets. Recently, structured state space models (SSMs) have emerged, with a time complexity that scales sub-quadratically with the context length. SSMs provide an efficient parameterization of sequence modeling relative to attention-based architectures. Given the success of Mamba and Mamba-2 in natural language, we designed BarcodeMamba, a performant and efficient foundation model for DNA barcodes in biodiversity analysis. We conducted a comprehensive ablation study on the impacts of self-supervised training and tokenization methods, and compared both versions of Mamba layers in terms of expressiveness and their capacity to identify "unseen" species held back from training. Our study shows that BarcodeMamba has better performance than BarcodeBERT even when using only 8.3% as many parameters, and improves accuracy to 99.2% on species-level accuracy in linear probing without fine-tuning for "seen" species. In our scaling study, BarcodeMamba with 63.6% of BarcodeBERT's parameters achieved 70.2% genus-level accuracy in 1-nearest neighbor (1-NN) probing for unseen species.;
Your verdict: Yes

Example 3:
Paper title: An ADHD Diagnostic Interface Based on EEG Spectrograms and Deep Learning Techniques;
Paper abstract: This paper introduces an innovative approach to Attention-deficit/hyperactivity disorder (ADHD) diagnosis by employing deep learning (DL) techniques on electroencephalography (EEG) signals. This method addresses the limitations of current behavior-based diagnostic methods, which often lead to misdiagnosis and gender bias. By utilizing a publicly available EEG dataset and converting the signals into spectrograms, a Resnet-18 convolutional neural network (CNN) architecture was used to extract features for ADHD classification. The model achieved a high precision, recall, and an overall F1 score of 0.9. Feature extraction highlighted significant brain regions (frontopolar, parietal, and occipital lobes) associated with ADHD. These insights guided the creation of a three-part digital diagnostic system, facilitating cost-effective and accessible ADHD screening, especially in school environments. This system enables earlier and more accurate identification of students at risk for ADHD, providing timely support to enhance their developmental outcomes. This study showcases the potential of integrating EEG analysis with DL to enhance ADHD diagnostics, presenting a viable alternative to traditional methods.;
Your verdict: Yes

*Table 14.* Few-shot learning samples in IPI task.

**IPI Few-shot Examples (Cont.)**

Example 4:

Paper title: Graph Neural Controlled Differential Equations For Collaborative Filtering; Paper abstract: Graph Convolution Networks (GCNs) are widely considered state-of-the-art for recommendation systems. Several studies in the field of recommendation systems have attempted to apply collaborative filtering (CF) into the Neural ODE framework. These studies follow the same idea as LightGCN, which removes the weight matrix or with a discrete weight matrix. However, we argue that weight control is critical for neural ODE-based methods. The importance of weight in creating tailored graph convolution for each node is crucial, and employing a fixed/discrete weight means it cannot adjust over time within the ODE function. This rigidity in the graph convolution reduces its adaptability, consequently hindering the performance of recommendations. In this study, to create an optimal control for Neural ODE-based recommendation, we introduce a new method called Graph Neural Controlled Differential Equations for Collaborative Filtering (CDE-CF). Our method improves the performance of the Graph ODE-based method by incorporating weight control in a continuous manner. To evaluate our approach, we conducted experiments on various datasets. The results show that our method surpasses competing baselines, including GCNs-based models and state-of-the-art Graph ODE-based methods.;

Your verdict: No

Example 5:

Paper title: Mechano-Bactericidal Surfaces Achieved by Epitaxial Growth of Metal-Organic Frameworks;

Paper abstract: Mechano-bactericidal (MB) surfaces have been proposed as an emerging strategy for preventing biofilm formation. Unlike antibiotics and metal ions that chemically interfere with cellular processes, MB nanostructures cause physical damage to the bacteria. The antibacterial performance of artificial MB surfaces relies on rational control of surface features, which is difficult to achieve for large surfaces in real-life applications. Herein, we report a facile and scalable method for fabricating MB surfaces based on metal-organic frameworks (MOFs) using epitaxial MOF-on-MOF hybrids as building blocks with nanopillars of less than 5 nm tip diameter, 200 nm base diameter, and 300 nm length. Two methods of MOF surface assembly, in-situ growth and ex-situ dropcasting, result in surfaces with nanopillars in different orientations, both presenting MB actions (bactericidal efficiency of 83% for E. coli). Distinct MB mechanisms, including stretching, impaling, and apoptosis-like death induced by mechanical injury are discussed with the observed bacterial morphology on the obtained MOF surfaces.;

Your verdict: No

*Table 15.* Few-shot learning samples in IPI task (Cont.).

**I3 Few-shot Examples**

Example 1:
Paper in Discipline 1:
- title: "Relation Between Retinal Vasculature and Retinal Thickness in Macular Edema"
- abstract: "This study has investigated the relationship of retinal vasculature and thickness for Macular Edema (ME) subjects. Ninety sets Fluorescein Angiograph (FA) Optical Coherence Tomography (OCT) 54 participants were analyzed. Multivariate analysis using binary logistic regression model was used to association between vessel parameters thickness. The results reveal feature i.e. fractal dimension (FD) as most sensitive parameter changes in associated with ME. Thus, indicating a direct which is caused due neovascular causing exudates, leakages hemorrhages, applications alternate modality detection"

Paper in Discipline 2:
- title: "An Image is Worth 16x16 Words: Transformers for Image Recognition at Scale"
- abstract: "While the Transformer architecture has become de-facto standard for natural language processing tasks, its applications to computer vision remain limited. In vision, attention is either applied in conjunction with convolutional networks, or used replace certain components of networks while keeping their overall structure place. We show that this reliance on CNNs not necessary and a pure transformer directly sequences image patches can perform very well classification tasks. When pre-trained large amounts data transferred multiple mid-sized small recognition benchmarks (ImageNet, CIFAR-100, VTAB, etc.), Vision (ViT) attains excellent results compared state-of-the-art requiring substantially fewer computational resources train."

Your verdict: Yes
Your reason: A novel work can combine transformers with two distinct methods that evaluate the quality of retinopathy", Confidence score: 92

Example 2:
Paper in Discipline 1:
- title: "Channelrhodopsin-2, a directly light-gated cation-selective membrane channel"
- abstract: "Microbial-type rhodopsins are found in archaea, prokaryotes, and eukaryotes. Some of them represent membrane ion transport proteins such as bacteriorhodopsin, a light-driven proton pump, or channelrhodopsin-1 (ChR1), recently identified light-gated channel from the green alga Chlamydomonas reinhardtii . ChR1 ChR2, related microbial-type rhodopsin C. , were shown to be involved generation photocurrents this alga. We demonstrate by functional expression, both oocytes Xenopus laevis mammalian cells, that ChR2 is directly light-switched cation-selective channel. This opens rapidly after absorption photon generate large permeability for monovalent divalent cations. desensitizes continuous light smaller steady-state conductance. Recovery desensitization accelerated extracellular H + negative potential, whereas closing decelerated intracellular expressed mainly under low-light conditions, suggesting involvement photoreception dark-adapted cells. The predicted seven-transmembrane $\alpha$ helices characteristic G protein-coupled receptors but reflect different motif Finally, we may used depolarize small simply illumination."

Paper in Discipline 2:
- title: "Shannon capacity of signal transduction for multiple independent receptors, DESIGN AND IMPLEMENTATION OF VISIBLE LIGHT COMMUNICATION SYSTEM IN INDOOR ENVIRONMENT"
- abstract: "Cyclic adenosine monophosphate (cAMP) is considered a model system for signal transduction, the mechanism by which cells exchange chemical messages. Our previous work calculated Shannon capacity of single cAMP receptor; however, typical cell may have thousands receptors operating in parallel. In this paper, we calculate transduction with an arbitrary number independent, indistinguishable receptors. By leveraging prior results on feedback receptor, show (somewhat unexpectedly) that achieved IID input distribution, and n times receptor. Visible Light communication (VLC) using White Light Emitting Diode (LED) is a promising technology for next generation communication for short range, high speed wireless data transmission."

*Table 16.* Few-shot learning samples in I3 task.

---

**I3 Few-shot Examples (Cont.)**

---

Your verdict: Yes

Your reason: An interdisciplinary paper can aim to use channelrhodopsin-2 (ChR2), a biomolecule, as a receiver to design a light-based communication system, which is a work related to engineering.

Confidence score: 85

Example 3:

Paper in Discipline 1:

- title: "A General Adaptive Dual-level Weighting Mechanism for Remote Sensing Pansharpening"

- abstract: "Currently, deep learning-based methods for remote sensing pansharpening have advanced rapidly. However, many existing methods struggle to fully leverage feature heterogeneity and redundancy, thereby limiting their effectiveness. We use the covariance matrix to model the feature heterogeneity and redundancy and propose Correlation-Aware Covariance Weighting (CACW) to adjust them. CACW captures these correlations through the covariance matrix, which is then processed by a nonlinear function to generate weights for adjustment. Building upon CACW, we introduce a general adaptive dual-level weighting mechanism (ADWM) to address these challenges from two key perspectives, enhancing a wide range of existing deep-learning methods. First, Intra-Feature Weighting (IFW) evaluates correlations among channels within each feature to reduce redundancy and enhance unique information. Second, Cross-Feature Weighting (CFW) adjusts contributions across layers based on inter-layer correlations, refining the final output. Extensive experiments demonstrate the superior performance of ADWM compared to recent state-of-the-art (SOTA) methods. Furthermore, we validate the effectiveness of our approach through generality experiments, redundancy visualization, comparison experiments, key variables and complexity analysis, and ablation studies. Our code is available at https:github.comJie-1203ADWM."

Paper in Discipline 2:

- title: "Secure Semantic Communication With Homomorphic Encryption"

- abstract: "In recent years, Semantic Communication (SemCom), which aims to achieve efficient and reliable transmission of meaning between agents, has garnered significant attention from both academia and industry. To ensure the security of communication systems, encryption techniques are employed to safeguard confidentiality and integrity. However, traditional cryptography-based encryption algorithms encounter obstacles when applied to SemCom. Motivated by this, this paper explores the feasibility of applying homomorphic encryption to SemCom. Initially, we review the encryption algorithms utilized in mobile communication systems and analyze the challenges associated with their application to SemCom. Subsequently, we employ scale-invariant feature transform to demonstrate that semantic features can be preserved in homomorphic encrypted ciphertext. Based on this finding, we propose a task-oriented SemCom scheme secured through homomorphic encryption. We design the privacy preserved deep joint source-channel coding (JSCC) encoder and decoder, and the frequency of key updates can be adjusted according to service requirements without compromising transmission performance. Simulation results validate that, when compared to plaintext images, the proposed scheme can achieve almost the same classification accuracy performance when dealing with homomorphic ciphertext images. Furthermore, we provide potential future research directions for homomorphic encrypted SemCom."

Your verdict: No

Your reason: The two papers are not related to each other. The first paper focuses on remote sensing pansharpening, while the second paper discusses secure semantic communication with homomorphic encryption. There is no clear interdisciplinary connection between them.

Confidence score: 90

---

*Table 17.* Few-shot learning samples in I3 task (Cont.).

---

**I2R Paper Comparison Prompt**

---

You are given a main paper and two candidate papers. Use the least reasoning effort to decide which one is better in forming a Interdisciplinary Research idea with the main paper.

After you provide your choice, provide a score from 0 to 100 to indicate your confidence level in the correctness of making this choice.

Keep in mind a good Interdisciplinary Research idea includes the following standards: * This research idea should be Interdisciplinary, whereas the idea stems from the combination of ideas from the two papers introduced above.

* The Interdisciplinary Research ideas should follow this definition:

Interdisciplinary Research is a mode of research that integrates information, data, techniques, tools, perspectives, concepts, and/or theories from two or more disciplines or bodies of specialised knowledge to advance fundamental understanding or to solve problems whose solutions are beyond the scope of a single discipline or area of research practice.

* This research idea should be feasible, whereas the hypothesis is not purely theoretical and can be validated by experiments.

* This research idea should be novel, whereas it is not only rare but also ingenious, imaginative, or surprising.

* This research idea should be useful, whereas it applies to the stated problem and is effective at solving the problem.

Note: The confidence level indicates the degree of certainty you have about your verdict and is represented as a percentage. For instance, if your confidence level is 80, it means you are 80 percent certain that your answer is correct and there is a 20 percent chance that it may be incorrect.

Main paper title: %s;
Main paper abstract: %s;

Paper 1 title: %s;
Paper 1 abstract: %s;

Paper 2 title: %s;
Paper 2 abstract: %s;

Requirements:
- Use the least thinking effort to make your decision.
- Use the template (in this format, with no markdown and lines separated by '
n') to provide your answer.
Choice: A simple answer containing either "Paper 1" or "Paper 2".
Confidence score: A numeric score ranging from 0 to 100

---

*Table 18.* Paper comparison prompt in I2R task.

---

**Abstract Generation Prompt**

---

Use the most important of the key concepts of both papers below to generate a new abstract that is novel and interdisciplinary. You should not summarize the abstracts, you should write a novel research idea and ignore the structure (you should not cite paper 1 or paper 2), focusing on content and integration. After writing your abstract, provide a score from 0 to 100 to indicate your confidence level in the quality of your abstract as an interdisciplinary research idea.

Keep in mind a good Interdisciplinary Research idea includes the following standards:

* This research idea should be Interdisciplinary, whereas the idea stems from the combination of ideas from the two papers introduced above.

* The Interdisciplinary Research ideas should follow this definition: Interdisciplinary Research is a mode of research that integrates information, data, techniques, tools, perspectives, concepts, and/or theories from two or more disciplines or bodies of specialised knowledge to advance fundamental understanding or to solve problems whose solutions are beyond the scope of a single discipline or area of research practice.

* This research idea should be feasible, whereas the hypothesis is not purely theoretical and can be validated by experiments.

* This research idea should be novel, whereas it is not only rare but also ingenious, imaginative, or surprising.

* This research idea should be useful, whereas it applies to the stated problem and is effective at solving the problem.

Note: The confidence level indicates the degree of certainty you have about your verdict and is represented as a percentage. For instance, if your confidence level is 80, it means you are 80 percent certain that your answer is correct and there is a 20 percent chance that it may be incorrect.

Think carefully to write a novel abstract that clearly states the objective of the paper, how the two ideas will be integrated, and the expected results.

Note: The confidence level represents how confident you are in the quality of your abstract as an interdisciplinary research idea, expressed as a percentage. For example, if your confidence level is 80, it means you are 80 percent certain your abstract is good and there is a 20 percent chance it may be flawed.

Use the template (in this format, with no markdown and lines separated by '
n') to provide your answer.

Abstract: The abstract you wrote using the ideas from the two given papers.

Reason: A detailed sentence with less than 100 words that describes the reasons to form the new idea and concepts extracted from each paper to make the idea of the abstract above, focusing on the integration reason.

Confidence score: A numeric score ranging from 0 to 100

Paper in Discipline 1:
%s

Paper in Discipline 2:
%s

*Table 19.* Abstract Generation Prompt used in the human evaluation on IDR idea quality.

| **Abstract Rewrite Prompt** |
| --- |
| Read the following paper and rewrite its abstract.
Requirements:
* Keep length similar to the original abstract.
* You should not change its logic.
* Directly output your rewritten abstract in the answer section below.

Paper abstract: %s

Answer: |

*Table 20.* Abstract Rewrite Prompt used in the data augmentation.

| **Reason Generation Prompt** |
| --- |
| Read the definition of valid Interdisciplinary Research (IDR) below and write reason why the following two paper can or cannot form a valid IDR idea.

Definition of good IDR:
* This research idea should be Interdisciplinary, whereas the idea stems from the combination of ideas from the two papers introduced above.
* The Interdisciplinary Research ideas should follow this definition:
Interdisciplinary Research is a mode of research that integrates information, data, techniques, tools, perspectives, concepts, and/or theories from two or more disciplines or bodies of specialised knowledge to advance fundamental understanding or to solve problems whose solutions are beyond the scope of a single discipline or area of research practice.
* This research idea should be feasible, whereas the hypothesis is not purely theoretical and can be validated by experiments.
* This research idea should be novel, whereas it is not only rare but also ingenious, imaginative, or surprising.
* This research idea should be useful, whereas it applies to the stated problem and is effective at solving the problem.

Requirements:
* Structure your reasoning based on good IDR standards and refer to the label below. * You should emphasize the logic in your explaination why the ideas of two paper is likely to fail / succeed.
* The length of the reasoning should be less than 100 words.
* Directly output your reason in the reason section below.

Paper 1 abstract: %s
Paper 2 abstract: %s
Can paper 1 or paper 2 form IDR? Label: %s

Reason: |

*Table 21.* Reason generation prompt for negative samples in data augmentation.

**Key Idea Integration Comparison**

---

**Paper Title**: A lightweight Convolutional Neural Network based on U shape structure and Attention Mechanism for Anterior Mediastinum Segmentation

**LLM I3 Reasoning**: Combining AI-powered lesion segmentation with lung cancer screening guidelines can improve diagnosis accuracy.

**Annotated I3 Reasoning**: In this paper, we introduce a U-shaped structure network to segment AM. Two attention mechanisms were used for maintaining long-range dependencies and localization. In order to have the potential of Multi-Head Self-Attention (MHSA) and a lightweight network, we designed a parallel MHSA named Wide-MHSA (W-MHSA).

---

**Full Abstract Generation Comparison**

---

**LLM Abstract:** This study proposes an interdisciplinary approach to investigating incidental anterior mediastinal nodular lesions on chest computed tomography (CT) scans in asymptomatic subjects, with a focus on early detection of lung cancer and thymic epithelial tumors. By integrating a UNet-based neural network with attention mechanisms for automatic detection of anterior mediastinal nodular lesions, and guidelines for lung cancer screening from the American Cancer Society, this research aims to develop a novel computer-aided detection system. The system will analyze low-dose chest CT scans to identify high-risk individuals and detect incidental nodular lesions, which can be potential thymic epithelial tumors or lung cancer. The expected outcome of this study is to improve the accuracy of early detection and diagnosis of lung cancer and thymic epithelial tumors, and to provide valuable insights into the prevalence and characteristics of incidental nodular lesions in asymptomatic subjects.

**True Abstract:** To automatically detect Anterior Mediastinum Lesions (AMLs) in the Anterior Mediastinum (AM), the primary requirement will be an automatic segmentation model specifically designed for the AM. The prevalence of AML is extremely low, making it challenging to conduct screening research similar to lung cancer screening. Retrospectively reviewing chest CT scans over a specific period to investigate the prevalence of AML requires substantial time. Therefore, developing an Artificial Intelligence (AI) model to find location of AM helps radiologist to enhance their ability to manage workloads and improve diagnostic accuracy for AMLs. In this paper, we introduce a U-shaped structure network to segment AM. Two attention mechanisms were used for maintaining long-range dependencies and localization. In order to have the potential of Multi-Head Self-Attention (MHSA) and a lightweight network, we designed a parallel MHSA named Wide-MHSA (W-MHSA). Maintaining long-range dependencies is crucial for segmentation when we upsample feature maps. Therefore, we designed a Dilated Depth-Wise Parallel Path connection (DDWPP) for this purpose. In order to design a lightweight architecture, we introduced an expanding convolution block and combine it with the proposed W-MHSA for feature extraction in the encoder part of the proposed U-shaped network. The proposed network was trained on 2775 AM cases, which obtained an average Dice Similarity Coefficient (DSC) of 87.83%, mean Intersection over Union (IoU) of 79.16%, and Sensitivity of 89.60%. Our proposed architecture exhibited superior segmentation performance compared to the most advanced segmentation networks, such as Trans Unet, Attention Unet, Res Unet, and Res Unet++.

---

*Figure 14.* Qualitative comparison sample on both idea integration reasoning and full abstract generation using `llama-3.3-70b`, both under zero-shot setting.

