# OpenReview forum: "IDRBench: Understanding the Capability of Large Language Models on Interdisciplinary Research"
_ICML.cc/2026/Conference — ICML 2026 regular_

### Official Review · Reviewer_Y6sg · 2026-03-07

**Soundness:** 2
**Presentation:** 3
**Significance:** 2
**Originality:** 3
**Overall Recommendation:** 4
**Confidence:** 4

**Summary:**

The paper introduces IDRBench, a benchmark to evaluate LLMs on cross disciplinary research ability. It defines three tasks including IDR paper identification, idea integration verification, and idea recommendation ranking, then reports results across multiple models using classification and ranking metrics.

**Compliance With Llm Reviewing Policy:**

Affirmed.

**Final Justification:**

My concerns have been addressed, I adjust my score to weak accept.

**Key Questions For Authors:**

1.If negatives are built by random cross discipline sampling, how often do they include genuinely integrable pairs that were simply not labeled as positives, and what is the estimated false negative rate.
2.Since pretraining exposure to PB and PC is treated as beneficial, how do you separate content recall from input grounded integration in the closed book setting.
3.With two annotators and only consensus positives retained, what are the agreement statistics and how does consensus filtering affect the difficulty distribution of the final triples.

**Limitations:**

The impact statement remains high level and does not substantively engage with potential risks such as misleading scientific assistance, citation contamination, or misuse in academic misconduct. Domain coverage is limited and the merging of disciplines into broader categories, together with an emphasis on more distant field combinations, may shift the task distribution and reduce benchmark representativeness and comparability.

**Strengths And Weaknesses:**

strengths：
1.The task design is clear and actionable, decomposing cross disciplinary research ability into identification, two paper integration verification, and multi candidate ranking for progressive diagnosis.
2.The positive examples are defined with an interpretable structure using key cited paper pairs plus an explicit integration statement, which helps separate mere citation from genuine integration.


weaknesses：
1.For I3 and I2R, negatives largely come from random cross discipline pairing with a 1 to 10 class imbalance, which can shift the task toward field distance or topical similarity cues rather than true feasibility of cross disciplinary integration.
2.The benchmark is framed as closed book, yet it also treats potential pretraining exposure as a feature and assumes titles and abstracts can trigger recall of paper content, making the evaluation closer to memorization and associative retrieval than input grounded integration.
3.The positive triple set is relatively small at about 335, and although dual annotation is used, agreement and dispute resolution are not reported in much detail, which may introduce some selection bias in the finalized set.

---

> ### Author Rebuttal · Authors · 2026-03-31
>
> ## Comment
> Class imbalance between negative and positive samples
>
>
> ## Response
>
> We appreciate your comment. For I3, we adopt a 1:10 class imbalance to better reflect a real-world scenario, considering the relatively low prevalence of IDR papers in the general literature, as confirmed by our analysis both in Section 3.2.1 in our paper and related work [8]. To verify whether setting a 1:10 ratio between positive and negative samples will shift topical similarity, we include additional analysis based on I3 subset 1 (with the negative instances selected randomly) in our dataset. Specifically, we extract the discipline plus subdiscipline labels from positive samples (335 pairs) and negative samples (3350 pairs) respectively, obtain the topical embeddings using sentence transformers, compute pair-wise discipline distance measured by cosine similarity (higher means more similar), and record the distribution. Our results show an average discipline cosine similarity of 0.267 (std = 0.118) in positive samples and 0.299 (std = 0.091) in negative samples, indicating that the class imbalance has little effect in hinting models to make decisions, meanwhile highlighting the importance of human evaluation. In fact, it is common for the positive set to include more diverse papers. For the negative set, however, there are both similar papers that do not meet the IDR definition and more distinct papers whose contexts are incompatible for interdisciplinary integration.
>
> For I2R, we also investigate whether the ranking results are influenced by the semantic similarity between cited paper PB and each candidate in the paper list L, rather than the true IDR idea combination. In other words, we analyze whether LLMs have developed an awareness that helps them distinguish IDR integration solely from paper similarity. Table 22 reports Kendall's τ correlation between the rankings produced by the LLM and those based on contextual similarity (using SciBERT similarity). As shown in the table, the correlation ranges from approximately 0.1 to 0.2, indicating that contextual similarity does not strongly influence the LLM’s re-ranking decisions, further suggesting that LLMs are capable of handling basic IDR recommendation tasks without reliance on paper similarity.
>
> In summary, the supplementary analyses for the I3 and I2R tasks validate the robustness of our dataset to ensure the authenticity of the LLMs' reasoning capabilities in IDR tasks.
>
> |  Model Name   | Kendall Correlation |
> | --- | -------- |
> |  Chatgpt-4o-mini  | 0.196  |
> |  Chatgpt-o3-mini   | 0.180  |
> |  Gemini-2.0-flash  | 0.220  |
> |  Llama-3.3-70B-instruct   | 0.211  |
> |  Claude-3.7-sonnet  | 0.249  |
> |  Deepseek-r1   | 0.194  |
>
> Table 22: Kendall correlation among I2R ranking and similarity scores of seed and candidate papers
>
> [8] Boyack, K. W. et al: https://akjournals.com/view/journals/11192/64/3/article-p351.xml
>
>
> ### Comment:
> Integration content separation in the close-book setting
>
> ## Response:
>
> Thank you for the comment. It is worth noting that, in our proposed triplet, PB and PC are the papers combined or integrated to create a new paper PA, and PA does not participate in pretraining corpus. Thus, the closed-book setting is defined as the model using only PB and PC as input (even if it has some prior memory of them) to generate PA, without any additional external information or intervention to support the integration.
>
> ## Comment:
>
> Annotator Consensus Statistics
>
> ## Response:
> Thank you for raising this point. It is worth noting that the proposed dataset is not easy to construct. Our human annotators are required to read the paper PA and identify the main cited papers PB and PC, including their core content. For agreement and dispute resolution, we emphasize that we include only instances where annotators reached agreement: i.e. we only retain a paper as a positive sample when the same paper is marked as positive twice by the annotators.
>
> ## Comment:
>
> Estimation of false negative rate in I3 samples
>
> ## Response:
>
> Thank you for raising this point. We remark that for the I3 task our dataset presents two distinct dataset: subset 1 uses random sampling across disciplines and subset 2 is not random; it defines negative instances as cases where papers PB and PC belong to the same discipline and subdiscipline, thereby not satisfying the IDR definition. But considering specifically the subset 1, we conduct additional human evaluations on the False Negative Rate (FNR) among the negative samples. Specifically, we randomly sampled 70 pairs and asked 3 annotators the feasibility of IDR combination. With an agreement rate of 85.5%, and an average FNR of 5.8% (we consider the rating of 4 and above as a false negative sample), it helps to enhance the validity of false pairs. In fact, even for Subset 1, we argue that IDR integrations are less frequent and more sparse compared to non-IDR papers, which is supported by our observations on arXiv and by related work [8].

---

> > ### Author Rebuttal · Reviewer_Y6sg · 2026-04-03
> >
> > My concerns have been addressed, I adjust my score to weak accept.

---

### Official Review · Reviewer_rLn2 · 2026-03-11

**Soundness:** 2
**Presentation:** 2
**Significance:** 2
**Originality:** 3
**Overall Recommendation:** 3
**Confidence:** 4

**Summary:**

This paper studies the capability of large language models to support interdisciplinary research. The authors introduce a benchmark consisting of three tasks: Paper Identification, Idea Integration, and Idea Recommendation, aiming to evaluate whether LLMs can assist researchers in discovering relevant literature and synthesizing cross-disciplinary ideas. The benchmark is constructed primarily from arXiv papers and includes expert annotations for certain components. Experimental results compare multiple LLMs across the proposed tasks.

**Compliance With Llm Reviewing Policy:**

Affirmed.

**Final Justification:**

Thank you to the authors for the detailed rebuttal and additional experiments. I appreciate the effort to address my concerns.

My main concern is still the verifiability of the benchmark, especially for the Idea Integration and Idea Recommendation tasks. While I agree that open-ended and subjective tasks can still be meaningful to study, for a benchmark it is important that the evaluation signal be sufficiently reliable, reproducible, and easy to interpret. In my view, the current evidence still does not fully establish this. The additional human studies help demonstrate that the generated ideas can sometimes be useful, but they do not fully resolve the underlying issue that these tasks involve a very large and subjective solution space, making objective evaluation inherently difficult.

The newly added results provide stronger evidence that the observed phenomenon (i.e., relatively weaker or less-aligned models performing better on this benchmark) is not an isolated case, but rather a more consistent trend. This is an interesting observation that, in my view, is worthy of further analysis. Accordingly, I increase my score.

Overall, I appreciate the motivation and believe the topic is important and interesting. However, due to the remaining concerns about benchmark validity and evaluation reliability, I do not feel sufficiently convinced to recommend acceptance.

**Key Questions For Authors:**

1. Can the authors provide an explanation for why LLaMA 3 achieves substantially stronger performance than other models on certain tasks?
It would be helpful to understand whether this is due to dataset characteristics, prompt design, or evaluation methodology.

2. Why was arXiv chosen as the primary data source instead of peer-reviewed conference papers?
The authors mention concerns about data leakage, but this issue could potentially be mitigated by selecting papers from conferences after certain cutoff dates. Although arXiv provides discipline labels, the paper also states that the final labels are annotated by experts, which somewhat reduces the advantage of arXiv metadata. Using peer-reviewed papers might provide higher-quality and more reliable ground-truth signals.

**Limitations:**

This paper doesn't explicitly discuss the limitations. Please refer to weakness part.

**Strengths And Weaknesses:**

**Strength**

1. The paper is generally easy to follow. The task definitions and benchmark construction are clearly described, making the evaluation pipeline understandable.

2. Studying the interdisciplinary research capability of LLMs is an important and under-explored direction. As LLMs are increasingly used as research assistants, evaluating their ability to connect ideas across disciplines is a meaningful problem that deserves further investigation.

**Weakness**

1. The main concern is the verifiability of the benchmark. The benchmark includes three tasks: Paper Identification, Idea Integration, and Idea Recommendation. Among them, Paper Identification appears relatively verifiable because it relies on labeled data that can be objectively checked. However, Idea Integration and Idea Recommendation involve a very large space of possible answers and rely heavily on subjective judgment. Evaluating the correctness or quality of generated ideas in these tasks is therefore difficult and typically requires assessment by experienced domain experts. In addition, the paper uses citing papers as ground-truth signals for idea relationships. However, the dataset is sourced from arXiv, where papers are not necessarily peer-reviewed and the quality varies substantially. As a result, citations extracted from arXiv papers may not reliably represent valid or high-quality idea integration. This raises further concerns about the reliability of the benchmark. For a benchmark, data quality and verifiability are fundamental, and the current task design does not sufficiently guarantee these properties.

2. The evaluated models appear outdated relative to the current state of LLMs. The experiments primarily evaluate models such as Gemini 2, LLaMA 3, and DeepSeek-V3. However, more recent and significantly stronger models—such as Gemini 3, Claude 4.6, and GPT-5—are now available and better represent the current frontier of LLM capabilities. Including these models would provide a more realistic assessment of the benchmark’s difficulty and relevance.

3. Some experimental results appear inconsistent with common expectations. For example, the results suggest that LLaMA 3 performs significantly better than other models on the I3 and I2R Set1 tasks, including OpenAI, Gemini, and Claude models. This observation seems inconsistent with common understanding of current model capabilities as well as results reported in other benchmarks. Such anomalies raise questions about whether the benchmark reliably measures the intended abilities.


Minor: The url in the abstract is a placeholder.

---

> ### Author Rebuttal · Authors · 2026-03-31
>
> ## Comment:
> Verifiability of benchmark
>
> ## Response:
> Thank you for the comment. We acknowledge the subjectivity and nuances inherent in evaluating idea integration. To mitigate this challenge, our I3 and I2R tasks utilize controlled settings with positive and negative instances to ensure reliability (Sections 3.2.2 and 3.2.3 in paper).
>
> To further validate quality, we conduct an extra evaluation among researchers in an academic institution across nine major disciplines and apply IDRBench tasks to their own published work. I2R identified pairs with interdisciplinary collaboration potential, while I3 generated research proposals based on those pairs. These proposals were sent directly to the original authors for evaluation to assess the practical viability of the AI-generated ideas.
>
> We collect 80 proposal evaluations from the researchers (1-5 scale) across three criteria. Score distributions (1-2, 3, 4-5) were: Integration (34, 15, 31); Clarity (24, 16, 40); and Interest (37, 13, 30). Analysis reveals a polarized, bimodal distribution. While Clarity achieved the highest positive concentration (>50% at scores 4-5), Integration and Interest were concentrated at extremes, reflecting the highly subjective nature of interdisciplinary synergy.
>
> Consistent with Figure 5 in paper, the new study confirms that over 50% of ratings find the generated ideas useful. The application of both I3 and I2R tasks in both evaluations underscores their robustness across diverse research scenarios.
>
> ## Comment:
> Use of arXiv and human annotation
>
> ## Response:
> Thank you for this thoughtful comment. While dataset quality is fundamental, relying solely on peer-reviewed journals risks data leakage. Ideas often appear on ArXiv long before publication cycles conclude, rendering traditional sources potentially outdated and already present in model training sets.
>
> ArXiv provides vast interdisciplinary diversity across 155+ subdisciplines, minimizing bias and enabling precise assessment of conceptual distance. Furthermore, utilizing ArXiv aligns our methodology with established IDR frameworks [2,3,4] found in recent literature.
>
> To quantify how human annotation ensures quality, analysis of 335 positive samples shows 80.2% are published or under review, while 63.6% of remaining preprints have citations. This confirms ArXiv captures novel research that matures into formal publications. Because ArXiv metadata is often noisy, our expert human evaluation (Section 3.1) complements these sources to maximize quality and verifiability.
>
> [2] Raimbault, J. et al: https://arxiv.org/abs/2006.16399.
> [3] Muralidharan, B. et al: https://arxiv.org/abs/2408.04651.
> [4] Jin, Y. et al: https://arxiv.org/abs/2410.09510.
>
> ## Comment:
> Evaluation on newer models
>
> ## Response:
> Thank you for the comment. We have included the evaluation with more recent models.
>
> | | GPT-o4-mini | GPT-5 | qwen-2.5-72B-instr | qwen-3-32B-instr |
> |-|-|-|-|-|
> | IPI - zero shot | 0.630 | 0.646 | 0.571 | 0.527 |
> | IPI - 5 shot | 0.640 | 0.669 | 0.501 | 0.565 |
> | I3 - zero shot | 0.814  | 0.230 | 0.793 | 0.782 |
> | I3 - 5 shot | 0.822  | 0.262 | 0.860 | 0.746 |
>
> Table 21: Additional experiment results on newer models and results comparison between instruct models and chat models.
>
> ## Comment:
> Llama 3 performance clarification
>
> ## Response:
> We appreciate the reviewer’s concern. The I3 task differs from mainstream benchmarks (e.g. GSM8K, MMLU) by incorporating multifaceted factors inherent in creativity evaluations. Section 4 in our paper explores the optimism-pessimism trade-off: optimistic models (higher TPR, lower TNR) favor creative synthesis despite hallucination risks, while pessimistic models (lower TPR, higher TNR) avoid false positives but may reject valid creative integration. Figure 4-b illustrates these tendencies, identifying Claude Sonnet 4 and GPT-o4-mini as pessimistic, while Gemini 2.0 Flash and Llama 3.1 70B lean optimistic.
>
> Our analysis shows Llama 3.3 occupies a balanced middle ground, avoiding over-conservatism and excessive hallucinations. Its performance likely stems from conventional post-training (SFT + RLHF); unlike strictly aligned closed-source models, Llama 3.3’s less restrictive schema permits diverse idea integration while instruction tuning minimizes false positives. Similar trends appear in Qwen 2.5 Instruct, which also outperforms advanced models (Table 21). These findings are supported by literature [5,6,7]: [6] suggests heavy alignment hinders creative effectiveness, while [7] highlights Llama’s aptitude for creativity.
>
> In summary, our results do not imply Llama 3 is generally more capable than GPT, Claude, or Gemini. Instead, I3 and I2R focus on instruction following and controllable generation—specific creativity dimensions where Llama 3.3’s heavy post-training excels.
>
> [5] Mohammadi, B. et al: https://arxiv.org/abs/2406.05587.
> [6] West, P. et al: https://openreview.net/forum?id=vqN8uom4A1.
> [7] Zhao, Y. et al: https://doi.org/10.1007/s11633-025-1546-4

---

> > ### Author Rebuttal · Reviewer_rLn2 · 2026-04-03
> >
> > I thank the authors for their detailed rebuttal and additional experiments on recent models. However, my main concerns have not been sufficiently resolved.
> >
> > 1. I still find the Idea Integration and Idea Recommendation tasks to be highly unverifiable. Although the authors provide small-scale human evaluations suggesting that the pipeline can generate potentially useful research ideas, the core issue is that these tasks inherently involve a large and subjective solution space. In my view, such tasks are difficult to evaluate objectively, and verifiability is a fundamental requirement for a benchmark. The current evidence does not sufficiently address this concern.
> >
> > 2. I am also not convinced by the claim that LLaMA 3 is particularly strong at exploring research ideas. The cited work by West, P. et al. compares base and instruction-tuned models, which does not directly explain the observed phenomenon here. If the hypothesis is that stronger alignment reduces creativity, it would be important to test this more directly (e.g., by comparing models like LLaMA 2 vs. LLaMA 3 under controlled settings). Without such analysis, the current explanation remains inconclusive.
> >
> >
> > Based on the above considerations, my overall evaluation remains unchanged.

---

> > > ### Author Response · Authors · 2026-04-07
> > >
> > > **Response to Comment 1**
> > >
> > > We thank the reviewer for reviewing our paper.
> > >
> > > We respectfully disagree with the comment on the relationship between large solution spaces, verifiability, and contributions. In fact, many well-established tasks—such as text summarization, open-ended generation, dialogue systems, many HCI tasks, among others—work on inherently subjective and expansive solution spaces. Despite this, in the past few decades, these fields have achieved sustained and measurable progress through well-designed evaluation protocols, including reference-based metrics, model-based evaluation, and carefully controlled human assessments. The presence of a large solution space does not preclude verifiability; rather, it necessitates rigorous and structured evaluation design, which is an objective of our benchmark.
> > >
> > > Specifically in our context, the tasks follow real scenarios under IDR. The IDR (i.e., citing) papers are real papers written by human researchers, based on which we design our evaluation tasks. If verifiable refers to whether the results, methods, or claims can be independently checked and confirmed by others, our dataset has undergone a rigorous double-checked human annotation and evaluation (commonly applied in many LLM works for evaluation).  Human studies are only one component of the annotation and evaluation we performed. During annotation, for each datapoint, multiple researchers identified closely related papers, followed by assessing whether their combination constitutes a valid interdisciplinary integration. They also provide their rationales. The human evaluation is comparable to studies in many NLP tasks, considering the efforts required from the experts. The user-centered study involves 56 researchers. All answers included descriptions justifying the decisions. This provides further empirical support from real-world usage scenarios.
> > >
> > > Regarding false negative cases, where two papers result in a valuable IDR idea that has never been explored by any researchers yet, this will have little impact on our evaluation, because (1) our negative examples have been validated by annotators to exclude such cases; (2) the probability that this will happen during test will be very small (most random pairs of papers will be actual true negative cases). As shown in the paper, many LLMs are, instead, more prone to yield false positive predictions.
> > >
> > > Having said that, we do appreciate the comment. We will add the discussions into our paper. We believe that IDR benchmarks need to be established, even if the solution space is large and subjective. We humbly argue that IDRBench has taken a careful process towards establishing a rigorous and controllable entry point for evaluating LLMs within the broad and open-ended domain of interdisciplinary research.
> > >
> > > **Response to Comment 2**
> > >
> > > Thank you for the comment. We added additional comparison between different generations of Llama on the I3 subset 1 task showcased in Table 23. We have also double-checked the current implementation---note that the I3 task is straightforward---only asking models to answer yes/no.
> > >
> > > We due the observation that LIama 2 did not outperform LIama 3 on IDR tasks to the difference in generation ability between them (which is in turn due to the difference of their pre-training data, post-training alignment, as well as other factors). However, this does not undermine our observation that relatively weakly-aligned models retain a more balanced attitude towards idea integration results (i.e. not being overly pessimistic or being overly optimistic), where models like LIama 3-instruct, Qwen 2.5-instruct, and Qwen 3-instruct are compared within the same generational horizon (similar pre-training corpus, and similar tech stack for post–training, etc). Given the primary focus of our paper being benchmark-oriented, we leave this as future work.
> > >
> > >
> > > | | I3 Subset 1 0-shot | I3 Subset 1 Few-shot |
> > > |-|-|-|
> > > | llama-2-70B-chat | .416 | .525 |
> > > | llama-3-70B-instr | .645 | .836 |
> > > | llama-3.3-70B-instr | .814  | .822 |
> > > | qwen-2.5-72B-instr | .793  | .860 |
> > > | qwen-3-32B-instr | .782  | .746 |
> > > | claude-4-sonnet | .563  | .694 |
> > > | gpt-o4-mini | .495  | .492 |
> > > | gpt-5 | .230  | .262 |
> > >
> > > Table 23: Additional Results on I3 Subset 1 on different Llama models.
> > >
> > > Regarding the hypothesis that stronger alignment (or reasoning strategies) could reduce creativity, prior work [1,2] points that there is a complex trade-off over excessive training and creativity. Along the line, our paper showcase additional evidence. In the rebuttal phase, we provide the Qwen-2.5-72B-instr model evaluation, surpassing Llama 3 for I3 task with 0.86 of macro-F1 and outperforming intensive instruction alignment models, such as Gemini and ChatGPT. Overall, our observation suggests that more alignment does not necessarily improve creativity, while less training does not increase creativity.
> > >
> > > [1] https://arxiv.org/abs/2601.00747
> > >
> > > [2] https://arxiv.org/abs/2406.05587

---

### Official Review · Reviewer_dHKa · 2026-03-12

**Soundness:** 4
**Presentation:** 3
**Significance:** 3
**Originality:** 3
**Overall Recommendation:** 5
**Confidence:** 3

**Summary:**

This work presents benchmark tasks and corresponding datasets for assessing large language models’ capability in interdisciplinary research. They first select interdisciplinary work from arxiv and then construct datasets for benchmark tasks. Three tasks include: (1) IDR paper identification (IPI) task, which is to judge whether the given paper is IDR research, (2) IDR Idea Integration (I3) task, which predicts whether a given pair of two papers fits in the IDR research criteria, (3) IDR Idea Recommendation (I2R) task, which compares two candidate papers to judge which one is a better fit with the target, main paper.  The authors also compare and report the performances of 10 models (5 reasoning models and 5 non-reasoning models).

**Compliance With Llm Reviewing Policy:**

Affirmed.

**Final Justification:**

Thanks for the response. I will keep my score.

**Key Questions For Authors:**

In 5.2. Error Analysis on I3 Task, the authors describe that lower performance of reasoning models on I3 task might be the reasoning-creativity tradeoffs. It came as a bit of a surprise to me. Why do the authors think interdisciplinary research is a creative task?

**Limitations:**

The limitations are not specified, nor is the potential negative societal impact of their work.

**Strengths And Weaknesses:**

I enjoyed reading this work. The manuscript is well-written. Each decision in the study, such as the arXiv paper inclusion time window, the combination of prevalent disciplines, and so on, is clearly explained. I also like that the authors involve human experts in creating datasets and validating the research output.
The authors design tasks to measure large language models' capabilities in interdisciplinary research using real-world data, which I think makes the work novel compared to prior work.

---

> ### Author Rebuttal · Authors · 2026-03-31
>
> We would like to thank all reviewers for acknowledging our work as novel, well-motivated, technically sound, and manuscript writing easy to follow.
>
> ## Comment
> > “In 5.2. Error Analysis on I3 Task, the authors describe that lower performance of reasoning models on I3 task might be the reasoning-creativity tradeoffs. It came as a bit of a surprise to me. Why do the authors think interdisciplinary research is a creative task?”
>
> ## Response
> Thank you so much for bringing up this question. Our stance of IDR being a type of creative task derives from its widely accepted definition mentioned in Section 3 in our paper: “integrate information.. from two or more disciplines ..to solve problems whose solutions are beyond the scope of a single discipline.” Thus, we believe that IDR by nature encourages solving problems using creativity to go beyond the scope of a single discipline, which demands drawing insightful analogies between domains that represent distinct disciplines or knowledge. As is argued in [1], the ability of making analogies across domains is regarded as the core of human thinking that drives creativity. Therefore, we believe IDR aligns with this underlying notion heavily and thus formulate our aforementioned stance.
>
> [1] Hofstadter, D. et al: Surfaces and Essence: Analogy as the Fuel and Fire of Thinking.

---

> > ### Author Rebuttal · Reviewer_dHKa · 2026-04-05
> >
> > Thanks for the response. I will keep my score.

---

### Decision · Program_Chairs · 2026-04-30

**Decision:**

Accept (regular)

**Comment:**

This paper introduces IDRBench, a benchmark for evaluating LLMs on interdisciplinary research through three tasks: paper identification, idea integration, and idea recommendation, constructed using arXiv data with expert annotations. All reviewers agree the problem is important and the task design is clear. Two reviewers give positive recommendations (accept / weak accept), highlighting novelty and practical relevance. However, Reviewer rLn2 raises major concerns about the verifiability and reliability of the I3 and I2R tasks, arguing that their subjective and open-ended nature makes objective evaluation difficult.

During the rebuttal, the authors provide additional experiments, human studies, and clarifications, which substantially improve confidence in the dataset construction and evaluation. While concerns about subjectivity remain, they are partially mitigated and aligned with similar challenges in established NLP benchmarks.